



# Environmental controls on surf zone injuries on high-energy beaches

Bruno Castelle[1,2], Tim Scott[3], Rob Brander[4], Jak McCarroll[3], Arthur Robinet[5], Eric Tellier[6,7,8], Elias de Korte[9], Bruno Simonnet[8], Louis-Rachid Salmi[6,7,10]

[1] CNRS, UMR EPOC, Univ. Bordeaux, Pessac, France
[2] Univ. Bordeaux, UMR EPOC, Pessac, France
[3] Coastal Processes Research Group, School of Biological and Marine Sciences, University of Plymouth, Plymouth, UK
[4] School of Biological, Earth and Environmental Sciences, UNSW Sydney, Australia
[5] Bureau de Recherche en Géologie Minière, Orléans, France
[6] INSERM, ISPED, Centre INSERM U1219 Bordeaux population health research, Bordeaux,
[7] Univ. Bordeaux, ISPED, Centre INSERM U1219 Bordeaux population health research, Bordeaux, France
[8] CHU de Bordeaux, Pôle Urgences adultes, SAMU-SMUR, Bordeaux, France
[9] Institute for Marine and Atmospheric Research, Department of Physical Geography, Faculty of Geosciences, Utrecht University, Utrecht, Netherlands
[10] CHU de Bordeaux, Pôle de Santé publique, Service d'information médicale, Bordeaux, France

*Correspondence to*: Bruno Castelle (bruno.castelle@u-bordeaux.fr)

**Abstract.** The two primary causes of surf zone injuries (SZIs) worldwide, including fatal drowning and severe spinal injuries, are rip currents (rips) and shore-break waves. SZIs also result from surfing and body boarding activity. In this paper we address the primary environmental controls on SZIs along the high-energy meso-macrotidal surf beach coast of SW France. A total of 2523 SZIs recorded by lifeguards over 186 sample days during the summers of 2007, 2009 and 2015 were combined with measured and/or hindcast weather, wave, tide and beach morphology data. All SZIs occurred disproportionately on warm sunny days with low wind likely because of increased beachgoer numbers and hazard exposure. Relationships were strongest for shore break and rip related SZIs and weakest for surfing related SZIs, the latter being also unaffected by tidal stage or range. Therefore the analysis focussed on bathers. Shore-break related SZIs disproportionately occur during shore-normal incident waves with average to below-average wave height (significant wave height $Hs$ = 0.75-1.5 m) and around higher water levels and large tide range when waves break on the steepest section of the beach. In contrast, rip related drownings occur disproportionally near neap low tide, coinciding with maximized channel rip flow activity, under shore-normal incident waves with $Hs$ > 1.25 m and periods mean wave period longer than 5 s. Additional drowning incidents occurred at spring high tide, presumably due to small-scale swash rips. The composite wave and tide parameters proposed by Scott et al. (2014) are key controlling factors determining SZI occurrence, although the risk ranges are not necessarily transferable to all sites. Summer beach and surf zone morphology is highly interannually variable, which is critical to SZI patterns. The upper beach slope can vary from 0.06 to 0.18 between summers, resulting in low and high shore-break related SZIs, respectively. Summers with coast-wide highly (weakly) developed rip channels also result in widespread (scarce) rip related drowning incidents. With life risk defined in terms of the number of people exposed to life threatening hazards at a beach, the ability of morphodynamic



models to simulate primary beach morphology characteristics a few weeks/months in advance is therefore of paramount importance to predict the primary surf-zone life risks along this coast.

# 1 Introduction

Sandy surf beaches are an attractive environment for a variety of recreational activities globally including sunbathing, swimming and wading, bodyboarding and surfing (West, 2005). However, surf beaches can also be dangerous environments to those who choose to enter the water due to potentially powerful wave conditions, strong currents and the risk of collision and impact injury involving surf craft. Although global statistics are not available, annual fatal drownings on surf beaches worldwide are at least in the hundreds (Gilchrist and Branche, 2016).

It is well established that the leading cause of both fatal and non-fatal drowning incidents on surf beaches worldwide are strong, narrow seaward flowing rip currents (hereafter referred to as 'rips'). Rips often originate close to the shoreline and can extend well beyond the breakers (Brander and Scott, 2016). They are capable of transporting bathers of all swimming abilities offshore into deeper water, increasing the risk of drowning through panic and exhaustion (Brander et al., 2011; Drozdzewski et al., 2012; 2015). Annually, they are responsible for an estimated 57 and 21 fatal drownings each year on surf beaches in the United States and Australia alone (NOAA, 2017; Brighton et al., 2013). They are also considered to be the cause of the vast majority of lifeguard rescues and SZIs in most documented regions of the world (e.g. 81% in the US (USLA, 2015); 57.4% in Australia (Brighton et al., 2013); 34% UK (RNLI, 2017)). Although rips can form through a variety of driving mechanisms, and have different hydrodynamic behaviours, they are essentially driven by the action of breaking waves (Castelle et al., 2016a). One of the most common rip types worldwide flow through channels incised in nearshore sandbars (e.g. MacMahan et al., 2006; Houser et al., 2013; Winter et al., 2014). These channel rips exhibit considerable natural variability in terms of flow characteristics and behaviour (McCarroll et al., 2018) preventing the public promotion of any single simple and universal escape strategy for those who find themselves caught in one (Bradstreet et al. 2014; McCarroll et al., 2014, 2015; Castelle et al., 2016b).

More recently, attention has been given towards the threat posed to bathers by shore-break waves associated with plunging/dumping waves at the shorelines of steep beaches. The concentrated and intense wave impact can cause surf zone injuries (SZIs), including severe spine injuries, to those caught in the impact zone (Robles, 2006; Puleo et al., 2016). Shore-break waves are, on average, associated with a low proportion of surf zone rescues and injuries in most of existing studies (e.g. 8.5% of leading cause of rescue in Australia (Brighton et al., 2013); 13% in UK (RNLI, 2017)), although higher proportions can be found (44.6% of the SZIs on the coarser grained and steeper beaches of SW France in Castelle et al., 2018a). More locally, shore-break waves were found to be responsible for over 82% of the SZIs at the steep beaches of Ocean City, Maryland (Muller, 2018). Another group of SZIs, which has received somewhat less attention and has been systematically treated separately from incidents involving bathers, are injuries associated with surfing, body boarding and other surf craft. Such high-risk activity (Diehm and Armatas, 2004) can lead to a wide spectrum of sustained spinal pathologies (Dimmick et al., 2013;



Nathanson, 2013), sprains/strains, fractures, dislocations, as well as lacerations (Moran and Webber, 2013) to the surf craft users and other people in the water. Lacerations, which are caused by the sharp fin, tail or nose of the surfboard are the most common injuries sustained whilst surfing (Lowdon et al. 1983; Nathanson, 2013).

Several attempts have been made to correlate the occurrence of these surf zone hazards and, in some cases, resulting rescues

and SZIs with environmental conditions. In the case of rips, it is well established that channel rip activity is controlled by offshore wave conditions (e.g. Brander, 1999; MacMahan et al., 2005; Castelle et al., 2006; Austin et al., 2010; Bruneau et al., 2011; Winter et al., 2014) and tide elevation (Aagaard et al., 1997; MacMahan et al., 2006; Bruneau et al., 2009; Austin et al., 2014). Surf zone morphology is also critical to rip activity, with deeper rip channels resulting in more intense rips (Brander, 1999; Aagaard and Vinther, 2008; Moulton et al., 2017a; McCarroll et al., 2018). Given that rip speed is an effective proxy

for the physical hazard posed by rips (Moulton et al., 2017b), and that warm sunny days with low winds typically result in increased beach attendance and beachgoer exposure to hazards (Ibarra, 2011), rip risk predictors based on simple correlations between meteorological, oceanographic conditions and the incidence of rip related rescues were developed (Lushine, 1991; Lascody, 1998, Dusek and Seim, 2013).

Such simple risk predictors are challenged in high-energy meso- and macro-tidal environments. Large water level variations

deeply modulate rip flow and hazard (Austin et al., 2013) and breaker type (Wright and Short, 1984) and beachgoer exposure can be strongly reduced for large waves as high surf tends to discourage beachgoers from entering the surf zone. However, beachgoer exposure data are unavailable in most instances. Therefore, reported SZI incidents and/or rescues provide information on the full life risk signal (Stokes et al., 2017), where life risk can be defined in terms of the number of people exposed to life threatening hazards at a beach, and their vulnerability to those hazards (Kennedy et al., 2013):

$LifeRisk = Hazars * Exposure$ ,                                                                                          (1)

In contrast to rips, studies on environmental controls on shore-break related SZIs are scarce and show poor correlations. For example, along the Delaware coast of the United States, Puleo et al. (2016) did not find any statistically significant relation between shore-break related SZIs and environmental factors such as tidal stage and foreshore slope. Although plunging and/or shore-break waves are known to occur for long-period waves and/or steep beach slopes (Battjes, 1974), Puleo et al. (2016)

only noted that the highest injury rates were associated with moderate wave heights. They therefore suggested that shore-break related SZI rates are primarily related to human factors. Finally, to our knowledge there is no scientific contribution on the environmental controls on SZIs sustained whilst using various types of surf craft.

Understanding the environmental controls on life risk along the coast requires accurate SZI records combined with detailed wave, tide and weather data at the time of the incidents. Focussing on rips along the meso-macrotidal rip-channelled beaches

of Devon and Cornwall in SW UK, Scott et al. (2014) compared the average frequency distribution of key environmental parameters (e.g. wave height, tide elevation, wind speed) with those computed from the environmental parameters associated with each recorded rip incident. Differences between the distributions showed that high-risk high-exposure scenarios for bathers occur disproportionately around mean low water on days with below average wave height, long wave period, shore-



normal wave approach and light winds. Scott et al. (2014) further developed two composite wave and tide parameters for which thresholds were successful in discriminating between high life risk events. However, it is unclear if these composite parameters work for other beaches (globally) and if similar thresholds are found, and it is unknown if they can be useful for other SZIs such as those caused by shore-break waves or sustained whilst surfing.

Morphological beach state, which exerts a strong control on the hazard posed by rips (Wright and Short, 1984) and shore-break waves (Battjes, 1974), is constantly evolving on the timescales from days (storms) to seasonal to multiannual (e.g. Splinter et al., 2014, 2018; Pianca et al., 2015; Davidson et al., 2017; Dodet et al., 2019) owing to changes in wave regimes. However all existing SZI and rip-rescue studies have overlooked the influence on life risk at the beach by key morphological parameters, such as variations in rip channel depth or upper beach slope. The influence of interannual morphological changes

on interannual life risk variability at a given coast is largely unexplored. Finally, shore-break and rip SZIs and/or rescues have been systematically treated separately in previous work, despite the fact that they can co-exist at certain surf beaches (Castelle et al., 2018a).

The SW coast of France hosts 230 km of meso-macrotidal sandy beaches that are visited by millions of tourists each summer (Brumaud, 2016) and are considered to be surf beaches. Castelle et al. (2018a) showed that along this stretch of coast, a large

number of bather SZIs occur due to shore-break waves and channel rips, with a large number of additional SZIs sustained whilst surfing. The combination of large tide range and variable incident energetic wave climate and weather conditions result in complex and highly variable patterns of SZIs. The primary aim of this paper is to build upon the dataset of lifeguard reported incidents of SZIs used by Castelle et al. (2018a) by examining extensive environmental data, including local to large-scale morphological data, to address the primary environmental controls on the full life risk spectrum at a high-energy meso-

macrotidal coast. This study also aims at addressing the genericity of the two composite wave and tide parameters to discriminate between low and high life-risk conditions. This contribution is the first to study the environmental controls, including coastal morphology, on all primary causes and activity leading to SZIs, and has strong implications towards future beach safety management and education of beach users in SW France, and globally, particularly if the parameters can be generalised.

**2 Introduction**

The study area is the Gironde-Landes coast in SW France, which extends 230 km from the Adour estuary in the south to the Gironde estuary in the north (Fig. 1a). The coastline is interrupted only by the large-scale Arcachon inlet separating the Landes and Gironde coasts in the south and north, respectively (Fig. 1a). Beaches are relatively straight (Fig. 1b) and primarily composed of fine to medium quartz sand with a mean grain size varying from 200 to 400 μm increasing southwards (Lorin

and Viguier, 1987). They are all backed by high and wide coastal dunes (Tastet and Pontee, 1998), except along some of the coastal towns (Capbreton, Mimizan, Biscarrosse, Montalivet, Soulac and Lacanau, Fig. 1, Castelle et al., 2018b). Beaches are exposed to high-energy ocean waves generated in the North Atlantic Ocean coming from the WNW direction (Butel et al.,


2002; Castelle et al. 2017). Overall, the wave height very slightly increases (< 10%) southwards because of the narrowing continental shelf reducing the bottom friction, resulting in less energy dissipation of the incoming ocean waves (Castelle et al., 2018b). The coast is meso-macrotidal with the tidal range marginally increasing northwards (< 15%) owing to the widening continental shelf (Le Cann, 1990). Neap tidal range is typically smaller than 1.5 m with the highest astronomical tidal range

reaching approximately 5 m. The beaches are rèip-dominated (e.g., Bruneau et al., 2011) and mostly intermediate double-barred (Castelle et al., 2007). The outer bar is modally crescentic (Lafon et al., 2005; Almar et al., 2010) while the inner bar is predominantly rip-channelled (Fig. 1b). Rip spacing of the inner and outer bars is about 400 m and 700 m, respectively, and progressively decreases northwards (Castelle et al., 2007).

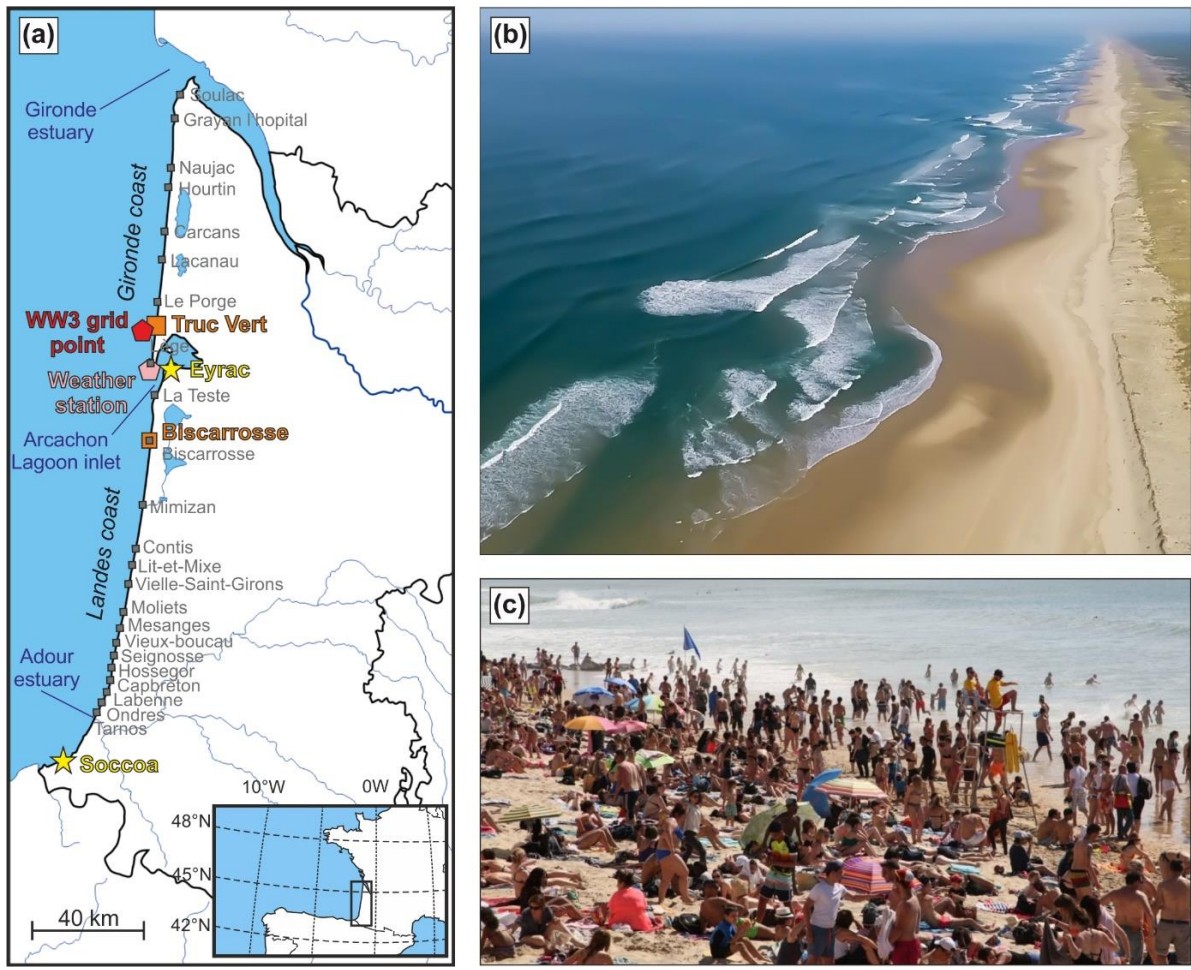

**Figure** Erreur ! Pas de séquence spécifié.**: (a) Location map of SW France showing the 230 km long sandy beach study location encompassing the Gironde and Landes between the Adour and Gironde estuaries. Grey squares indicate coastal municipalities where incident/injury report forms were gathered, yellow stars indicate tide gauges, and the red and pink polygons show the location of the WW3 model output grid point and weather station, respectively. (b) Aerial photograph of Truc Vert beach taken at low tide showing a landscape representative of the SW France beaches with deep rip channels and steeper upper beach slopes (Ph. V.**
**Marieu). (c) Crowded beach during summer at high tide with lifeguards supervising a bathing area delimited by 2 blue flags, typically extending no more than 100 m alongshore (Ph. J. Lestage).**



The wave climate and resulting beach morphology are strongly seasonally modulated. Offshore of Truc Vert, which is located approximately at the centre of the study region, the monthly-averaged significant wave height (Hs) peaks at 2.4 m during winter in January with a dominant W direction. The 99.5% significant wave height quantile $Hs_{99.5\%}$ is 5.6 m and severe storms $Hs > 8$ m can occur in winter. In contrast, summers are characterised by milder waves with a monthly mean $Hs$ of

approximately 1.2 m and a dominant W-NW direction (Castelle et al., 2017) although storms with $Hs > 3$ m can occur. In summer, beaches become more reflective with the inner bar often migrating from a transverse bar and rip state to a more reflective low tide terrace state according to the classification of Wright and Short (1984). Steeper upper beach slopes during summer also favour beach cusp development in the vicinity of the high tide mark, and beaches are usually steeper in the south owing to larger mean grain size (Lorin and Viguier, 1987). Summer beach morphology is also strongly variable from year to

year in relation to the strong interannual wave climate variability (Robinet et al., 2016; Dodet et al., 2019).

Beaches are characterised by strong rips primarily flowing through the inner-bar rip channels (Bruneau et al., 2011, Fig. 1b) with mean flow speeds reaching 1 m/s even in relatively low wave energy conditions (< 1 m wave height; Castelle et al., 2016b). Rip flow is tide-modulated (through modulation of breaking wave patterns) with maximum rip activity typically between low and mid tide in summer wave conditions (Bruneau et al., 2009). In addition, near high tide, waves can break close

to the shoreline on the steepest beach sections creating hazardous shore-break conditions.

The SW coast of France also hosts some of the finest surfing beaches in Europe, attracting many surfers. During busy summer months, large numbers of surfers of all ability, including surf schools, expose themselves to hazards related to surfing activity. Beaches are patrolled by lifeguards during the summer months of July and August, with extended periods of patrols at the busiest beaches, with a designated and supervised bathing zone. However, given the length of the coast and the many remote

beach access paths through coastal dune tracks from inland carparks, many access points are situated on unpatrolled sections of beaches, kilometres away from any lifeguard presence. Approximately 4-5 million tourists, primarily from France and other European nations, come to the Gironde and Landes coasts each year to enjoy the beaches (Brumaud, 2016) and simultaneously exposing themselves to surf zone hazards (Fig. 1c). Thousands of SZIs are sustained each year of which over a thousand are documented by lifeguards in injury report forms (Castelle et al., 2018a). Overall, surf zone Injuries sustained range from mild

contusion to fatal drowning and include severe spinal injuries, wounds and dislocation (Castelle et al., 2018a).

## 3 Introduction

The dataset of lifeguard reported incidents of SZIs in SW France presented in Castelle et al. (2018a) is combined with extensive environmental data to address the primary environmental controls and high-risk conditions leading to bather SZIs caused by shore-break waves and channel rips, and to SZIs sustained whilst surfing. The environmental dataset includes weather data,

local beach and large-scale coastal morphology, tide and wave hindcast.





## 3.1 Data

The SZI dataset collected in Castelle et al. (2018a) is used here for further analysis. Each time a medical incident occurs on the beach, a lifeguard responds to the scene to provide patient care and to potentially assist paramedics. An injury report form detailed in Castelle et al. (2018a) is filed for every incident. A total of 2523 SZIs over 186 sample days during the summers of

2007, 2009 and 2015 were collected and examined by Castelle et al. (2018a) who addressed the epidemiology of SZIs. Amongst all the data analysed in Castelle et al. (2018a), only date and time (< 10 min accuracy) of the incident; beach location; as well as categorically ticked activity (e.g. wading, surfing, bodyboarding); cause of injury (e.g. rip current, shore-break waves); and injury type (e.g. drowning, spine injury) are used herein. In addition, open-text field comments in the form were sometimes used for instance to understand the cause of incident (e.g. collision against someone else, or against their own

surfboard) as there is no such box to tick in the forms provided. In the case of a drowning incident, the drowning stage was also provided according to a 4-stage classification widely used in France (Menezes and Coasta, 1972; Dupoux et al., 1981): (1) exhaustion, but no sign of aspiration of water; (2) moderate respiratory impairment, anxiety; (3) altered consciousness, severe respiratory impairment or acute pulmonary oedema, tachycardia or hypotension; and (4) coma, respiratory or cardiac arrest.

Overall, Castelle et al. (2018a) showed that while rips cause the most severe injuries (drowning), a large number of SZIs, including severe spinal injuries, are caused by shore-break waves. Another large proportion of injuries are related to surfing and bodyboarding activity. Within the beachgoer cohort, surfers do not enter the water for the same reasons as regular bathers. Therefore, in the present contribution we discriminate injuries: (1) sustained while surfing or bodyboarding whatever the cause is, hereafter referred to as surfing related injuries (33.5% of the dataset, $n = 844$); (2) caused by rips, but excluding surfers and

bodyboarders (8.7% of the dataset, $n = 220$), hereafter referred to as rip related injuries, which essentially consist of fatal and non-fatal drowning incidents with rips causing 83% of the drowning incidents reported; (3) caused by shore-break waves, but excluding surfers and bodyboarders (41.7% of the dataset, $n = 1053$), which primarily result in contusion/sprain/dislocation (51%; $n = 537$) and spine injuries (29.8%, $n = 314$). For each SZI, the date, time and location (coastal municipalities in Fig. 1a) reported in the form were used to estimate the wave, tide and meteorological conditions at the time of the incident using

the environmental dataset described below.

## 3.2 Environmental data

In order to obtain a continuous time series of wave conditions, 6-hourly outputs from a wave hindcast (Boudière et al., 2013) based on the spectral wave model Wave Watch III (Tolman, 2014) were used. The hindcast was performed on an unstructured grid with a resolution increasing from 10 km offshore to 200 m near the coast, and has been extensively validated (Boudière

et al., 2013; Ardhuin et al., 2012). Given the weak gradient in wave conditions along the coast, the wave data at grid point 1.3232°W, 44.7374°N in approximately 30-m depth was used and considered as representative of the entire coast (Castelle et





al., 2018b). Using this dataset, each incident was associated with a significant wave height *Hs*, mean and peak wave periods *T02* and *Tp*, respectively, and wave angle *θ*.

A tidal component analysis of a 3-month time series of continuous, storm-free, Soccoa tide gauge data (Fig. 1) was performed to reconstruct a tide level time series (elevation every 10 min). The average phase lag between the Soccoa tide gauge and

beaches of all coastal municipalities (grey squares in Fig. 1) was estimated using tide charts from the Service Hydrographique et Océanographique de la Marine (France). Errors due to the (time-varying) phase lag and amplitude difference between real and predicted tide result in an estimated maximum error in tide elevation of 0.3 m at all sites. Using this dataset, each incident was associated with a tide elevation *η* and a daily tide range TR.

We used RADOME (Réseau d'Acquisition de Données d'Observations Météorologiques Etendu), which is an automatic

weather station network of Météo-France, that collects meteorological data at 554 stations across France. Hourly data of air temperature (*T*), mean wind speed (*W*) and insolation (*I*) collected at the Cap Ferret station (Fig. 1) were used to estimate weather conditions at the time of each incident. Given the relative homogeneity of the coast and weather patterns in the region, as well as the lack of other weather stations directly implemented on the coast, this data was assumed to be representative of the weather along the entire study area. The corresponding daily mean values ($\acute{T}$, $\acute{W}$ and $\acute{I}$) averaged during the patrolled hours

between 11h and 19h, were also computed.

The morphological state of the beaches was estimated primarily using the topographic data collected at Truc Vert beach located in the south of the Gironde coast (Fig. 1). From April 8 2005 to present, topographic surveys have been performed at spring low tide every 2 or 4 weeks with an alongshore coverage that progressively increased from approximately 300 m in 2005 to approximately 1500 m since 2012 (Castelle et al., 2017). SPOT and Sentinel satellite large-scale images taken at low tide,

which therefore show the intertidal inner-bar system were collected. This morphological dataset was used to address the overall differences in beach morphology, rip channel characteristics, and upper beach slope between the 3 summers.

Additional morphological data from Biscarrosse beach was also used. A field experiment was performed at Biscarrosse on June 13-17 2007, during which the topo-bathymetry was accurately surveyed along approximately 2000 m of beach (Bruneau et al., 2009, 2011). This morphology, which can be assumed to be representative of the beach morphology throughout the 2007

summer, was used to address wave transformation and wave-driven currents during that summer. For this purpose, an Xbeach model (Roelvink et al., 2009) was implemented on a 10 m x 10 m regular grid to address the influence of offshore wave conditions and tidal elevation on rip flow dynamics and resulting hazard. The model, which is depth-averaged and wave-group resolving, has been demonstrated to be suitable for this purpose even when default settings were used (Castelle et al., 2016b).

## 4 Results

### 4.1 Environmental controls

Fig. 2 shows time series of environmental factors and SZIs, discriminating rip related drownings and SZIs related to shore-break waves and surfing activity during the 2007 summer. This time series, which is representative in patterns of the two other





summers, clearly shows some complex relations between the occurrence of different SZIs and environmental factors. During this summer, the coast was exposed to a broad range of wave conditions, with $Hs$ and $Tp$ ranging approximately 0.4-4 m and 5-16 s, respectively, with waves coming from the SW to N, although predominantly from the WNW (Fig. 2a-c) resulting in highly variable amount of daily SZIs (Fig. 2j). The least temporally variable SZIs are those sustained whilst surfing, which occur almost throughout the summer (daily mean for the entire dataset $\acute{n} = 4.5$, standard deviation $\sigma = 3.3$). The only notable exception is for days with large daily-mean wind speed $\acute{W}$ (> 10 m/s), coming systematically from the SW-NW quadrant (not shown). These windy conditions systematically result in no, or a small number, of surfing related SZI(s) ($\acute{n} = 1.0$ with a maximum of 4, standard deviation $\sigma = 1.5$). This is likely due to strong onshore winds being typically associated with messy and choppy sea states that strongly deteriorate surfing wave quality and, in turn, the number of surfers entering the water. Rip ($\acute{n} = 1.2$, $\sigma = 2.1$) and shore-break ($\acute{n} = 6.0$, $\sigma = 8.0$) related injuries are much more variable in time with many days with no injury reported. These injuries tend to occur in clusters with particular days prone to rip related drowning incidents or shore-break related SZIs (Fig. 2h,i). Clearly, days with a large number of drowning incidents caused by rips are not necessarily associated with a large number of shore-break related injuries, and vice versa.

Considering the entire 3-year dataset, the number of rip and shore-break related daily injuries are weakly positively correlated ($r = 0.28$), similar to rip drowning incidents and surfing-activity related injuries ($r = 0.25$), while shore-break related injuries are more correlated with injuries resulting from surfing activity ($r = 0.43$). A possible explanation of this larger correlation is that amongst all the reported surfing related injuries ($n = 844$), 8.5% ($n = 72$) were reported to be caused by shore-break waves, as opposed to 0.6% ($n = 5$) caused by rips. This is in line with earlier studies showing that Injuries resulting from surfing and body boarding occur more frequently in shore-break conditions (e.g. Beratan and Osborne 1987; Chang et al., 2006).

Overall, there appears to be no single environmental factor controlling the occurrence of SZIs. Instead, it is possible that certain combinations of environmental controls are conducive to causing SZIs, and that these combinations depend on the type of SZI. To better understand these combinations promoting life-risk surf-zone conditions, the approach of Scott et al. (2014) was utilised whereby the environmental conditions during which incidents occurred were compared with the 'average' background conditions of the 2007, 2009 and 2015 summer seasons. The environmental parameters addressed here are: significant wave height $Hs$, mean wave period $T02$, peak wave period $Tp$, wave direction $\theta$, tide or water level $\eta$, daily tide range TR, daily averaged air temperature $\acute{T}$, insolation $\acute{I}$ and wind speed $\acute{W}$. The average frequency distribution of these parameters was compared with those computed from the environmental parameters associated with each recorded rip incident. Differences between the distributions therefore provides an indication of environmental conditions that may be driving SZIs (Scott et al., 2014).





**Figure 2: Time series during the 2007 summer of: (a) significant wave height *Hs*; (b) peak and mean wave periods *Tp* and *T02*, respectively; (c) wave direction *θ*; (d) water level *η* with respect to mean sea level; (e) daily-mean air temperature *T́*; (f) daily-mean hourly insolation *Í*; (g) daily-mean wind speed *Ẃ*; (h) distribution of rip related SZIs; (i) distribution of shore-break related SZIs; (j) distribution of surfing-activity related SZIs and total number of SZIs. In (a-g) the shaded green, red and blue vertical areas**





**indicate the days with the largest number of shore-break, surfing and rip related SZIs during the 2007 summer. In (a,b,c,e,f) the summer mean is indicated by the horizontal dashed line.**

Fig. 3 provides insights into the environmental controls on surfing related injuries in SW France. The frequency distributions

(Fn) of the wave parameters related to reported incidents show some differences from the average background distributions

with conditions of $Hs$ = 0.75-1.5 m, $T02$ = 6-8 s and $\theta$ = 290-300° being slightly over-represented by 10.5, 9.2 and 6.2%, with

60.3, 36.4 and 46.8% occurring within this range, respectively (Fig. 3a-c). In contrast, water level does not substantially affect

surfing related incidents as no clear pattern stands out (Fig. 3d), while daily tidal range TR shows a local peak near mid tide

range (MTR, Fig. 3e). Incidents occurring for low wind speed ($\acute{W}$ < 4 m/s), and for warm and sunny weather ($\acute{T}$ > 24°, $\acute{I}$ > 50

min/hr) were overrepresented by 10.1, 6.6 and 5.2%, with 32.2, 38.2 and 45.2% occurring within this range, respectively (Fig.

3f-h). Therefore high-risk surfing incident scenarios occur disproportionately on days with low to moderate (average to below

average), wave height, large (above average) wave period ($T02$ > 6s, and $Tp$ > 9 s, not shown), WNW incidence, and warm

sunny weather with low wind (below average), regardless of water level or daily tide range. As surfers can be considered as a

more experienced beachgoer community, the SZIs involving only bathers are assessed below, with shore-break and rip related

injuries investigated separately.

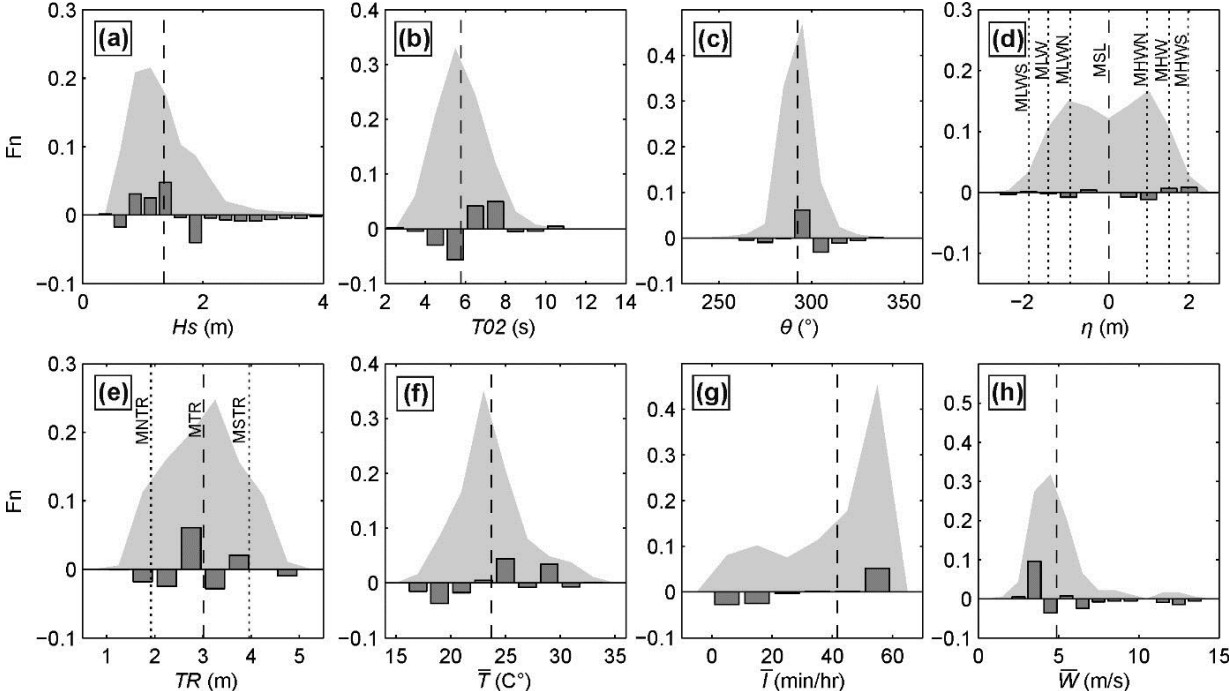

**Figure 3: Environmental controls on surfing-activity related SZIs: normalised frequency distributions Fn during the summers of 2007, 2009 and 2015 (light grey region), referred to as 'average' background distribution, of: (a) significant wave height $Hs$; (b) mean wave period $T02$; (c) wave direction $\theta$; (d) water level $\eta$; (e) daily tide range TR; (f) daily-mean air temperature $\acute{T}$ ; (g) daily-**

**mean hourly insolation $\acute{I}$ ; and (h) daily-mean wind speed $\acute{W}$. The dark grey bars show the difference between surfing related and 'average' background distributions and the vertical dashed lines indicate background means.**





Fig. 4 shows the same analysis, but for shore-break related injuries, and reveals more prominent, and sometimes different, patterns compared to surfing related injuries. The most salient difference is that tide plays a critical role with water level $\eta >$ 1.25 m (> MHWN) and daily tide range TR > 3.5 m (spring tides) being over-represented by 32.5 and 10.6%, with 13.5 and 27.4% occurring within this range, respectively (Fig. 4d,e). The influence of wave and weather conditions is also more pronounced than for surfing related injuries. The frequency distributions of the wave parameters related to reported incidents show some large differences from the average background distributions with conditions of $Hs = 0.75$-1.5 m (average to below-average), $T02 > 7$ s (above-average) and $\theta = 290$-300° being over-represented by 18.1, 6.8 and 8.6%, with 70.0, 16.0 and 46.8% occurring within this range, respectively (Fig. 4a-c). Incidents occurring at low below-average wind speed ($\acute{W} < 4$ m/s), and for warm and sunny weather ($\acute{T} > 24°$, $\acute{I} > 40$ min/hr) were overrepresented by 16.0, 26.5 and 14.6%, with 32.2, 38.1 and 62.9% occurring within this range, respectively (Fig. 3f-h). These are again in higher proportion than for surfing related injuries.

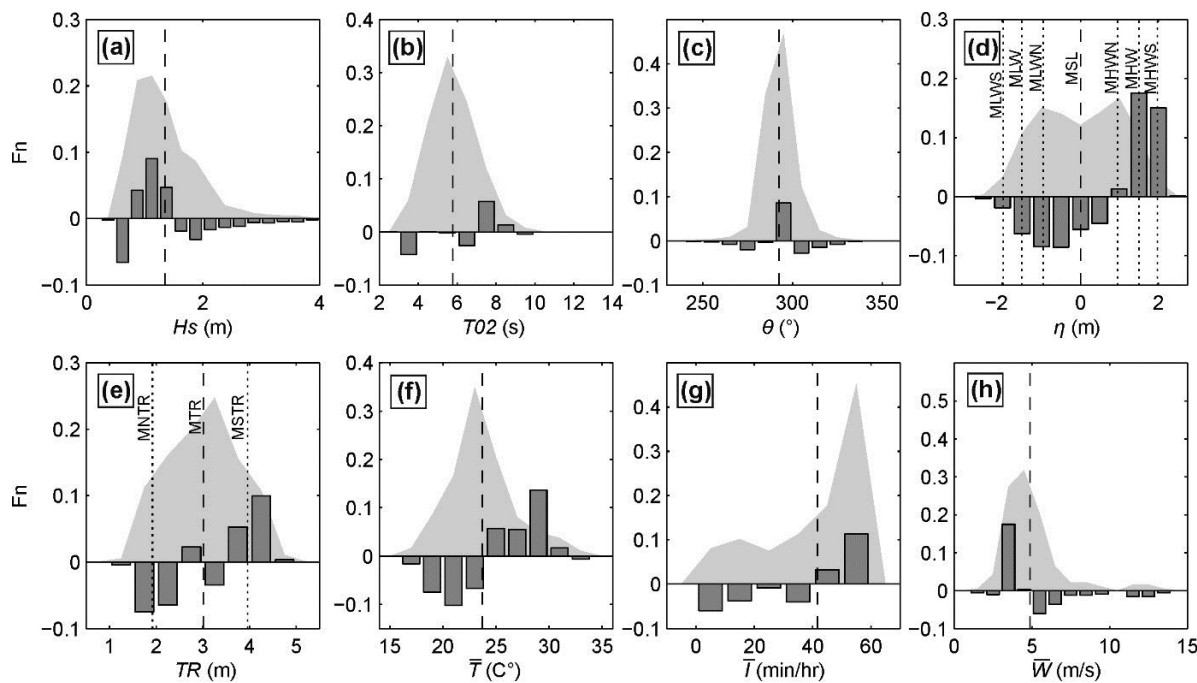

**Figure 4: Environmental controls on shore-break related SZIs: normalised frequency distributions Fn during the summers of 2007, 2009 and 2015 (light grey region), referred to as 'average' background distribution, of: (a) significant wave height _Hs_; (b) mean wave period _T02_; (c) wave direction θ; (d) water level η; (e) daily tide range TR; (f) daily-mean air temperature _Ť_; (g) daily-mean hourly insolation _Í_; and (h) daily-mean wind speed _Ŵ_. The dark grey bars show the difference between surfing related and 'average' background distributions and vertical dashed lines show the background mean.**

Fig. 5 shows the same analysis for rip related injuries. Similar to shore-break and surfing related injuries, rip related injuries occur disproportionally on warm, sunny days with light wind. Days with low wind speed ($\acute{W} < 4$ m/s) and warm, sunny weather ($\acute{T} > 24°$, $\acute{I} > 40$ min/hr) were over-represented by 7.2, 7.9 and 11.6%, with 32.2, 38.1 and 62.9% occurring within this range,




respectively (Fig. 5f-h). While this over-representation is less important than for shore-break related injuries, it is slightly more pronounced than for surfing related injuries. The control of waves on rip related SZIs contrasts with that on the other SZIs. Larger waves with more shore-normal incidence increase the likelihood of rip related injuries as significant differences from the average background distributions are found for conditions of $Hs > 1.25$ m (average to above-average) and $\theta = 280\text{-}290°$, with over-representation by 22.1 and 13.6%, and with 47.7 and 33.2% occurring within this range, respectively (Fig. 5a,c). Even more contrasting is the role of tide that shows complex patterns. Low tide levels $\eta < -0.75$ m and, to a lesser extent, extremely high tide levels $\eta > 1.75$ m are over-represented by 16.1 and 5.8%, with 29.3 and 2.7% occurring within this range, respectively (Fig. 5d). The daily tide range does not show any clear pattern (Fig. 5e). This complex control of tide on rip related injuries is discussed in Section 5.

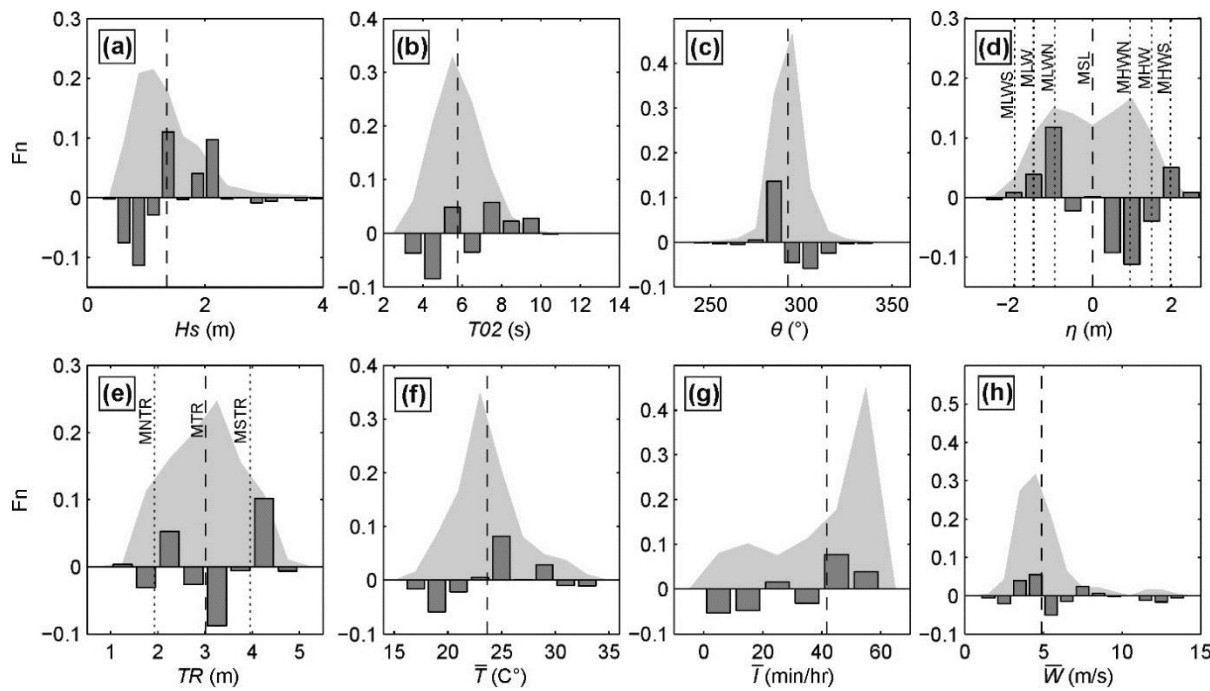

Figure 5: Environmental controls on rip related SZIs: normalised frequency distributions Fn during the summers of 2007, 2009 and 2015 (light grey region), referred to as 'average' background distribution, of (a) significant wave height *Hs*; (b) mean wave period *T02*; (c) wave direction *θ*; (d) water level *η*; (e) daily tide range TR; (f) daily-mean air temperature *Ť*; (g) daily-mean hourly insolation *Í* and (h) daily-mean wind speed *Ẃ*, with in all panels the dark grey bars showing the difference between the surfing related and the 'average' background distributions and the vertical dashed line showing the background mean.

## 4.2 High life-risk rip and shore-break days

As shown in Fig. 2, during the 2007 summer the number of SZIs per day was highly variable, particularly rip and shore-break related injuries, with some days having no incident reported and others with a large number of SZIs ($\acute{n} = 13.6$, $\sigma = 11.9$). Amongst the highest risk days, July 15 2007 had the highest number of SZIs (60) with incidents reported across 17 coastal municipalities, which is the most widespread day in the entire time series (daily average number of municipalities reporting incident(s) $\acute{m} = 7.4$, standard deviation $\sigma_m = 4.2$). During that day 68.3% ($n = 43$) and 13.3% ($n = 8$) were caused by shore-



break waves and rips, respectively. The second day with the largest number of SZIs (59 across 16 coastal municipalities), August 4 2009, was unusual because of very high shore-break life risk 81.4% ($n = 48$) SZIs caused by shore-break waves. This corresponds to the day with the largest number of shore-break related injuries in the entire dataset ($\acute{n} = 6.0$). Shore-break related SZIs were widespread along the coast as reported across 14 coastal municipalities ($\acute{m} = 3.9$, $\sigma_m = 3.6$). Of note, no rip

related injury occurred on August 4 2009. In contrast, August 5 2007 was characterized by less reported SZIs ($n = 37$), but by very high rip life risk. On this day, 40.5% ($n = 15$) of the reported incidents were drownings (maximum of the entire time series, $\acute{n} = 1.2$, $\sigma = 2.1$) across 8 different coastal municipalities ($\acute{m} = 1.0$, $\sigma_m = 1.5$), including 1 stage 3 and 2 deadly stage 4 incidents. Only 18.9% ($n = 7$) of the SZIs on August 5 2007 were caused by shore-break waves. While tide and wave conditions were very different for these two high risk days, both days were characterised by warm, sunny weather with light wind,

resulting in crowded beaches and high beachgoer exposure to surf-zone hazards (Fig. 6). The wave and tide controls during these two days are now addressed in detail.

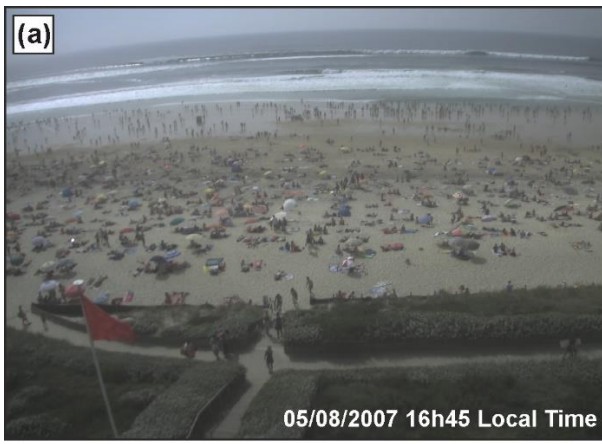

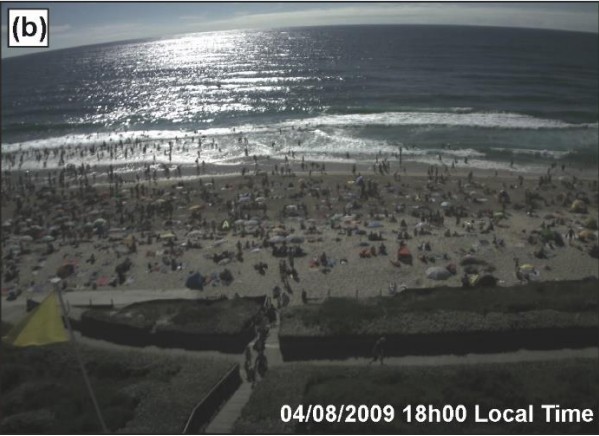

**Figure 6: Snapshots of Biscarrosse beach from a video station (Angnuureng et al., 2017) during the days with the largest number of SZIs related to: (a) rips on August 5 2007; and (b) shore-break waves on August 4 2009, with pictures taken approximately at the**

**time of the most hazardous conditions (Fig. 7; 10). Note that in panel (a) the red flag is hoisted meaning that bathing was forbidden all along the beach due to the presence of rips.**





Fig. 7 shows time series of SZIs, wave and tide conditions from two days prior to August 5 2007 when the largest number of rip related drownings in the entire dataset were reported. On August 3-4 2007, the coast was exposed to low-energy waves ($Hs \approx 0.5$ m, $Tp \approx 10$ s) with sunny weather ($\acute{I} = 60$ min/hr). The daily number of reported SZIs of 8 and 13 on August 3 and 4 2007, respectively, was relatively low or average compared to the mean ($\acute{n} = 13.6$), with two stage-2 drowning incidents on

the 4th. Given that the wave climate was similar for the two days, it is hypothesized that the increased air temperature on the 4th ($\acute{T} = 28.9°$) compared to the 3rd ($\acute{T} = 23.4°$) contributed to greater beach attendance and beachgoer exposure to surf-zone hazards and therefore a larger number of SZIs and higher life risk. August 5 2007 was still reasonably warm ($\acute{T} = 25.9°$) and sunny ($\acute{I} = 44.9$ min/hr) and also had high beach attendance (Fig. 6a). In contrast, on August 4-5 2007 (the two previous days) the coast was exposed to a shore-normal high-energy groundswell ($Hs \approx 2$ m, $Tp \approx 14$ s) with low tide levels occurring later

in the afternoon. The temporal distribution of rip related drowning incidents is clearly modulated by tidal stage, as incidents were most numerous between low and mid tide, decreased at low tide and did not occur from mid to high tide (Fig. 7a). Importantly, the number of drownings caused by rips on August 5 2007 could have been much larger as the red flag prohibiting bathing all along the beach was hoisted on many of the beaches in the afternoon (Fig. 6a), due to the large number of rescues and mass rescue interventions. This can also explain why rip related SZIs on August 5 2007 were reported across 8 coastal

municipalities. Rip related SZIs were the most widespread on July 25 2009 with 10 reporting coastal municipalities, under warm and sunny weather, shore-normal moderate energy waves ($Hs \approx 1.4$ m, $Tp \approx 9$ s) and spring low tide occurring in mid-afternoon. However, on July 25 2009 only 12 non-fatal drowning incidents (4 stage 1 and 8 stage 2) were reported, corresponding to smaller rip life risk than on August 5 2007 (no stage 1, 12 stage 2, 1 stage 3 and 2 fatal drownings).




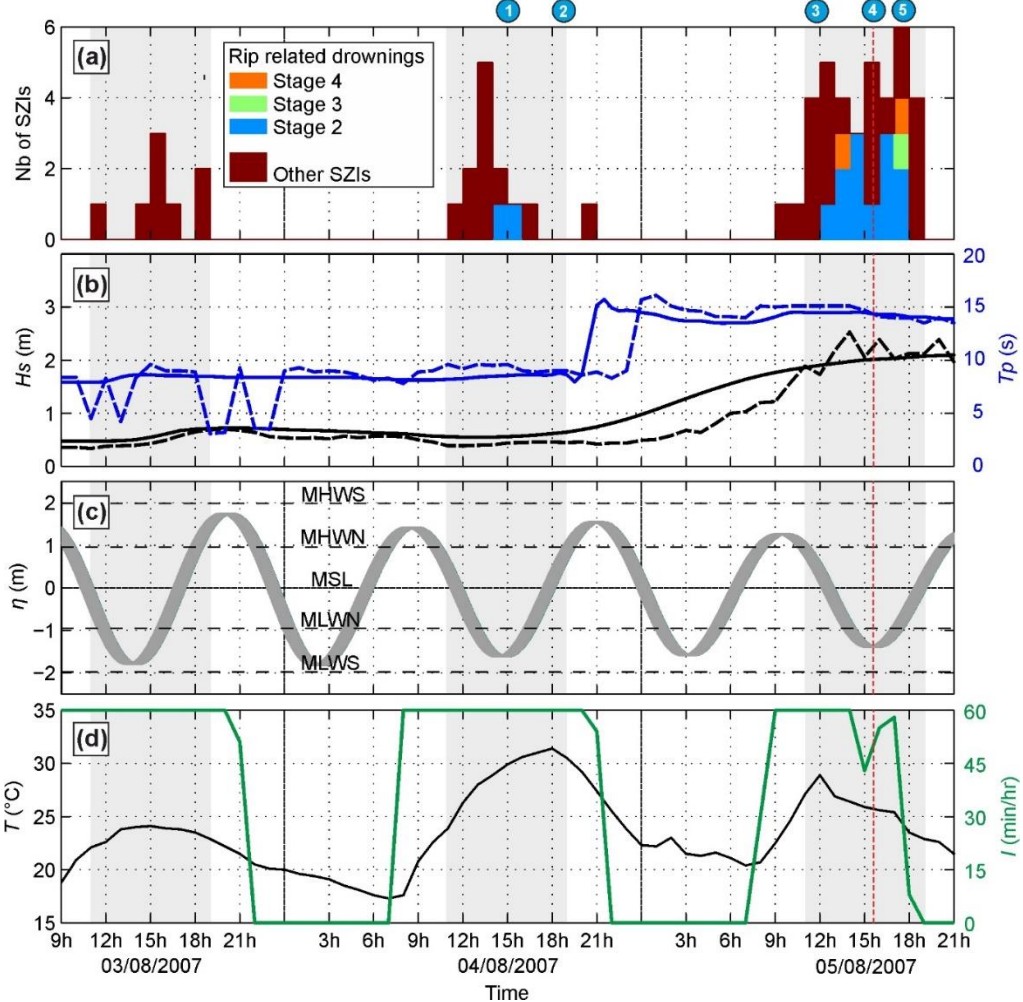

**Figure 7: Time series on August 3-5 2007, with August 5 having the largest number of rip related drownings in the entire dataset, showing: (a) distribution of SZIs with indication of rip related drowning and drowning stage coloured; (b) significant wave height *Hs* and peak wave period *Tp* from the model (solid) and from the Candhis directional wave buoy in 50-m depth facing Truc Vert which worked only intermittently during the three summer seasons; (c) tide elevation *η*; (d) hourly air temperature *T* and insolation *I*. In all panels, the light grey areas show the lifeguard patrolled hours. In all panels the vertical red dashed line indicates low tide occurrence on August 5 2007. The blue circles on top of panel (a) indicate the timing of the wave-driven current simulations shown in Fig. 8b-f.**

To further explore the influence of wave and tide conditions on rip activity on August 4-5 2007, Fig. 8 shows modelled breaker and wave-driven current patterns (using the morphology surveyed in mid-June 2007 at Biscarrosse in Bruneau et al., 2009) at relevant times indicated by the blue circles in Fig. 7. At low tide (15h) on August 4 2007, when two moderate (stage 2) drowning incidents were reported (Fig. 9a), low- to moderate-energy rips (0.3-0.4 m/s mean offshore velocities) were predicted by the model to flow through a reasonably narrow surf zone (Event 1), corresponding to a moderate rip hazard. In contrast, 4 hours later between mid and high tide, there was no rip activity observed along the entire beach and in turn no rip hazard posed to bathers, which is consistent with the absence of reported rip related SZIs (Event 2). On August 5 2007 simulations show


that rips were active throughout the entire tide cycle for energetic shore-normally incident groundswell conditions. Breaker and wave-driven current patterns at 12h, 15h and 17h are shown in Fig. 8d-f (Events 3-5). The time slot with the largest number of reported rip related drownings (2 stage 2, 1 stage 3, 2 stage 4) occurred between 17h and 18h (Event 5, Fig. 8f), that is, between low and mid tide ($\eta \approx$ -0.9 m). Simulations show the presence of wide (200+m) breakers with high-energy rips (0.8-

5    1.2 m/s mean velocities) flushing the surf zone. These rips are associated with intense alongshore northerly and southerly feeder currents, locally reaching 1.5 m/s and running close to the shoreline (Fig. 8f). These flow patterns therefore have a high potential to rapidly transport unsuspecting bathers wading close to the shoreline into strong seaward flowing rips, which could take them 300 m offshore in less than 5 minutes. Similar patterns are simulated earlier in the day for similar water levels, but slightly milder wave conditions (Event 3, Fig. 8d). Rips were the most intense (1-1.5 m/s mean velocities) at low tide ($\eta$ = -

10    1.5 m, Event 4, Fig. 8e). However, feeder currents were, overall, less intense and/or run further offshore, which may explain why fewer drowning incidents were reported the same day for higher water levels (Fig. 8d,f).

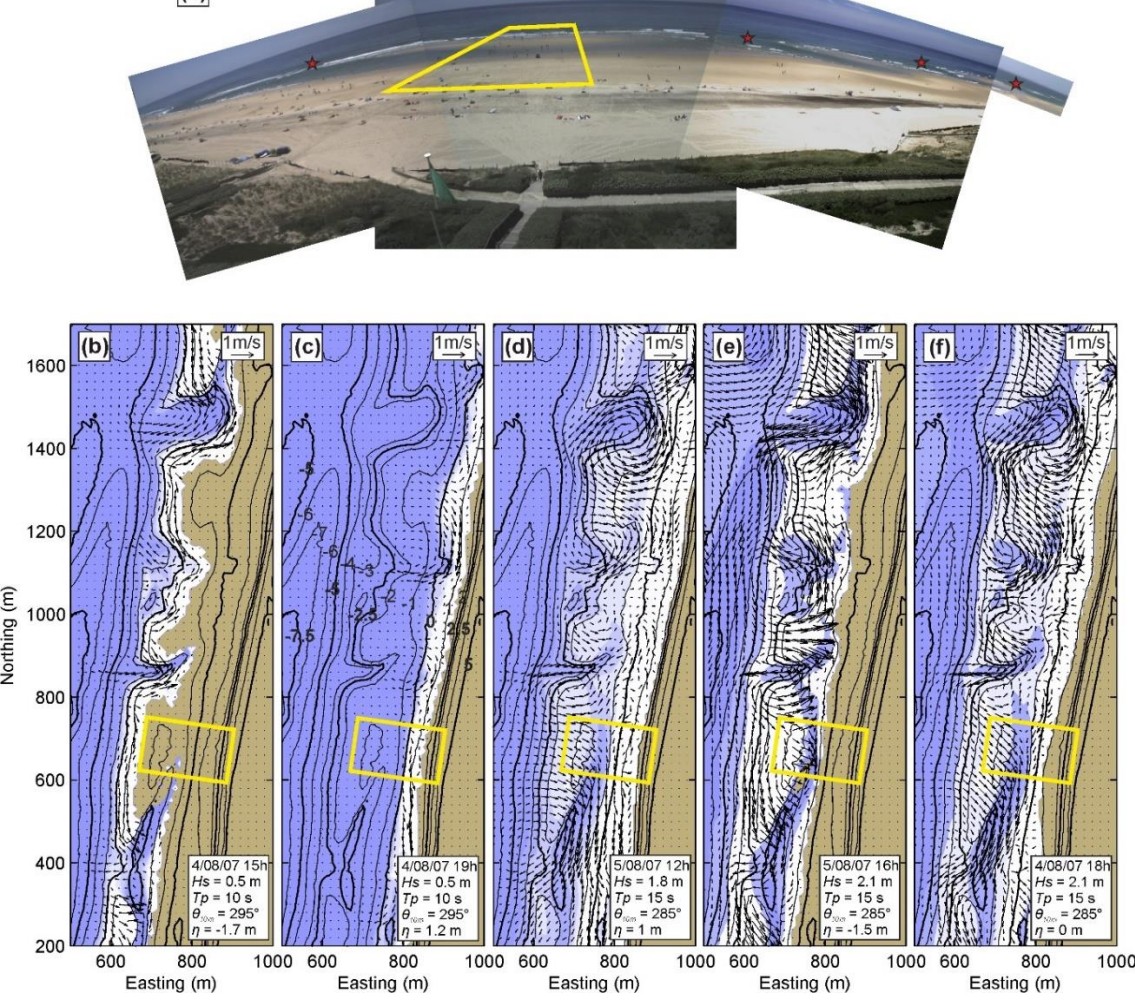



**Figure 8: (a) Biscarrosse beach on June 18 2007 with red stars showing rip channel location. Bottom panels show wave driven currents (arrows) and breaking wave patterns (white areas) simulated at the same beach during the 5 times on August 5 2007 indicated by the blue circles in Fig. 9a for the following wave and tide conditions: (b) $Hs$ = 0.5 m, $Tp$ = 10 s, $\theta_{10m}$ = 295°, $\eta$ = -1.7 m; (c) $Hs$ = 0.5 m, $Tp$ = 10 s, $\theta_{10m}$ = 295°, $\eta$ = 1.2 m; (d) $Hs$ = 1.8 m, $Tp$ = 15 s, $\theta_{10m}$ = 285°, $\eta$ = 1 m; (e) $Hs$ = 2.1 m, $Tp$ = 15 s, $\theta_{10m}$ = 285°, $\eta$ = -1.5 m; (f) $Hs$ = 2.1 m, $Tp$ = 15 s, $\theta_{10m}$ = 285°, $\eta$ = 0 m. In (b-f), the yellow box shows the preferred location of the swim between the flag area at Biscarrosse beach during most of the 2007 summer, and the topo-bathymetry is contoured in the background with contour elevation indicated in (c) with respect to mean sea level.**

Our results indicate that the most hazardous rip conditions were caused by a high-energy shore-normal groundswell ($Hs \approx 2$ m, $Tp \approx 14$ s), with low tide levels maximizing hazard in the afternoon, with risk also maximized due to high beachgoer afternoon exposure owing to warm and sunny weather. Such wave conditions are, however, quite rare in summer along the SW coast of France. Fig. 9 provides insight into the large-scale atmospheric patterns that resulted in these anomalous wave conditions. Five days prior to August 5 2007 a tropical depression intensified into Tropical Storm Chantal on July 31 while located at approximately 35°N in the West Atlantic. The storm tracked northeastward through an area of progressively colder water and cooler air and transitioned into an extratropical cyclone on August 1. It subsequently intensified to attain winds of near hurricane-force, and on August 3 the cyclone underwent a slowly weakening trend in the far northern Atlantic Ocean. This storm track drove high-energy waves funnelling towards the Atlantic coast of Europe with $Hs$ peaking at almost 10 m offshore of Ireland (Fig. 9b), coming from the $W$ and with $Tp$ exceeding 14 s (Fig. 9c).

The occurrence of high-energy groundswell events such as those on August 5 2007 is clearly linked with storm activity and trajectory during the hurricane season. The hurricane season officially begins on June 1 and ends on November 30 each year, with tropical storm and cyclone activity peaking in September. August and July are characterized by weaker tropical storm and cyclone activity. However, as shown by the trajectories of all tropical cyclones between 2005-2017 (light grey) and the July-August periods (dark pink) in Fig. 9a, there were quite a large number (12) of tropical cyclones that tracked northeastward across the north Atlantic at >40°N in July-August. These weather patterns have the potential to generate high-energy waves towards the beaches of SW France like those observed on August 5 2007. Of note, only 2 tropical cyclones tracking northeastward at >40°N were observed during the summers of 2007, 2009 and 2015, including Tropical Storm Chantal. The other one was long lasting (August 15-26 2009) Hurricane Bill which drove above-average wave height (1.5 m < $Hs$ < 1.5 m) and period (9 s < $Tp$ < 15 s) near shore-normally incident waves in SW France between August 20-28 2009. Although coinciding with below-average air temperature and partly cloudy weather at the end of the summer touristic season, likely resulting in decreased beachgoer numbers and hazard exposure, the period was characterized by 15 drowning incidents including 2 fatal incidents.





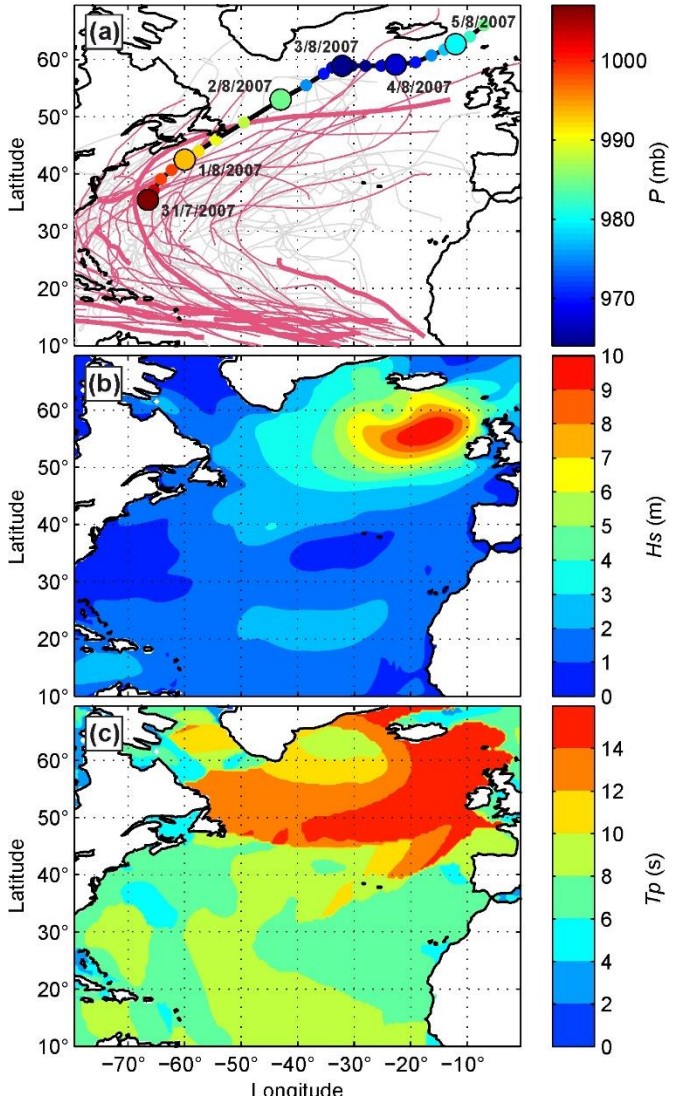

**Figure 9: Atmospheric conditions and resulting wave fields causing the most hazardous rip conditions of the dataset along the SW France beaches on August 5 2007: (a) Trajectory, daily position (large circles) and minimum central pressure of tropical cyclone Chantal superimposed onto all tropical cyclone trajectories over the period 2007-2017 (light grey) and limited to the July-August summer periods (dark prink) and only summers 2007, 2009 and 2015 (thick dark pink); (b) Significant wave height *Hs* and (c) peak wave period field on August 4, 2h00 local time.**

Fig. 10 shows the time series of SZIs, wave and tide conditions spanning the two days prior to August 4 2009 when the largest number of shore-break related SZIs in the entire dataset were reported. During the 3 days, incident wave conditions varied slightly with $0.7 < Hs < 1.5$ m, $7.5 < Tp < 10$ s and $280° < \theta < 300°$ with higher tide level occurring mid-afternoon, and high tide occurring increasingly late in the day. The major difference between the three days is the warmer and sunnier weather on August 4 2009. This likely resulted in high beach attendance and beachgoer exposure which, coinciding with higher tide levels, resulted in a large number of shore-break related injuries. Given that wave conditions were similar to modal summer wave





conditions in SW France, it suggests that maximized shore-break related risk occurs for moderate energy wave conditions, hot and sunny days and high tides that occur in late afternoon when beach attendance is maximized. This is further emphasized in Fig. 11, which shows that days with milder temperature ($\acute{T} < 22°$) or small ($Hs < 0.6$ m) or larger ($Hs > 2$ m) waves systematically result in a reduced occurrence of shore-break related SZIs. For intermediate wave conditions ($0.6$ m $< Hs < 2$

5    m), the number of shore-break related SZIs increases with increasing daily maximum tidal elevation between 15h and 19h $\eta_{max}$ when beach attendance is maximized during warm and sunny days. For such intermediate wave conditions, high-risk shore-break conditions are clearly observed for $\eta_{max} > 1.25$ m and $\acute{T} > 22°$ (Fig. 11).

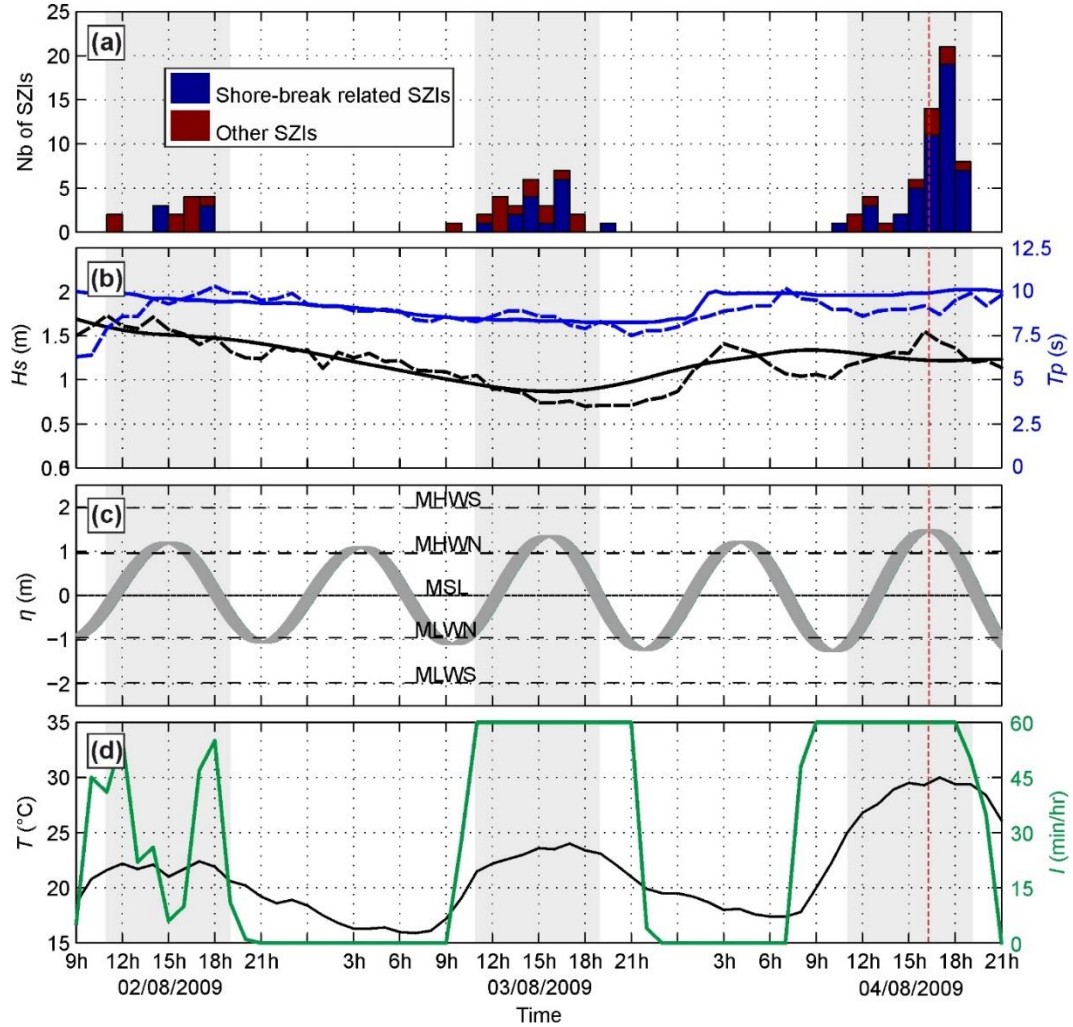

**Figure 10: Time series on August 2-4 2009, with August 4 having the largest number of shore-break related injuries in the entire
10    dataset, showing: (a) distribution SZIs with indication of shore-break related injuries; (b) significant wave height *Hs* and peak wave
period *Tp* from the model (solid) and from the Candhis directional wave buoy in 50-m depth facing Truc Vert which only worked
intermittently during the three summer seasons; (c) tide elevation *η*; (d) hourly air temperature *T* and insolation *I*. In all panels, the
light grey areas show lifeguard patrolled hours. In all panels, the light grey areas show the lifeguard patrolled hours. In all panels
the vertical red dashed line indicates low tide occurrence on August 4 2009.**




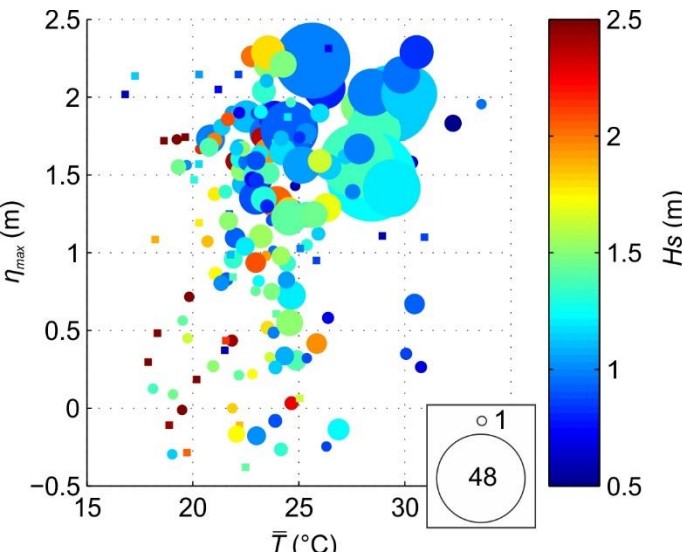

**Figure 11: Daily maximum tidal elevation $\eta_{max}$ between 15h and 19h versus daily mean air temperature averaged between 11h and 19h $\acute{T}$ with significant wave height averaged between 11h and 19h coloured and circle size proportional to the daily number of shore-break related injuries. Squares show days when there was no shore-break related injury.**

## 5 Discussion

Both shore-break and rip related SZIs involving bathers were found to occur disproportionately on warm sunny days with low winds, environmental conditions that are found to contribute to increased beach attendance and beachgoer exposure to hazards (e.g. Ibarra, 2011; Balouin et al., 2014). These conditions are also generally associated with normally incident waves with longer wave periods that have long been known to promote the occurrence of channel rips (e.g. Castelle et al., 2006). The high occurrence of rip related SZIs occur during such conditions in SW France and are consistent with those reported along the west coasts of Devon and Cornwall in SW UK (Scott et al., 2014). However, given that breaker type and the onset of shore-break waves is not known to be affected by angle of wave incidence (Battjes, 1974), it is somewhat surprising that shore-break related SZIs also occur disproportionately with near shore-normal incident waves. An explanation is that more shore-normal wave incidence in summer in SW France is typically associated with longer-period waves (Castelle et al., 2016b). The longer-period waves result in higher wave energy flux and favour the presence of plunging and collapsing waves for a given beach slope (Battjes, 1974).

Many studies have highlighted the important tidal control on channel rip activity with maximum rip flow speed often observed at lower tide levels (e.g. Aagaard et al., 1997; Brander, 1999; Brander and Short, 2001; MacMahan et al., 2006). This is also the case on the meso-macrotidal beaches of the SW coast UK (Austin et al., 2010; Scott et al., 2011, 2014) and SW France (Castelle et al., 2006; Bruneau et al., 2009, 2011), which helps explain why rip related SZIs in this study occur disproportionately around mean low water (Fig. 5d). Of note, a disproportionate number of rip incidents took place during spring high tide levels (Fig. 5d). A detailed inspection of open-text field comments in the corresponding injury report forms indicates that approximately 25% of these drowning incidents occurred because of small-scale swash rips (Russell and





McIntire, 1965; Dalrymple et al., 2011) referred to as *courant de ressac* by some lifeguards, which flow through the centre of beach cusps along the highest and steepest section of some beaches as wave uprush diverges at the cusp horns driving concentrated backwash streams (Masselink and Pattiaratchi, 1998). Tide also has a strong control on shore-break related SZIs, with injuries disproportionately occurring around higher water level (Fig. 4d). This is a logical consequence of shore-break

waves being favoured by steep beach slope, with the steepest sections of the beaches of SW France being near spring high tide (Castelle et al., 2017).

The impact of the daily tide range TR on SZIs is more complex (Fig. 4e and 5e), although shore-break related SZIs overall occur disproportionately for higher daily tide range TR (Fig. 4e). This can be explained by the fact that higher tide ranges result in a longer duration of high water level and sustained hazardous shore-break waves. In addition, Fig. 12a shows that for

larger daily tide range TR, daily maximum tide elevation η_max tends to occur during the patrolled hours in mid-late afternoon when beach attendance (exposure) is maximized. This can further explain the disproportionate number of shore-break incidents taking place for a large tide range. The same applies for the disproportionate number of rip related SZIs for higher tidal range TR (Fig. 5e), which is may be the signature of spring-high-tide swash rips. The other range of TR showing a disproportionate number of rip related SZIs is around mean neap tide range MNTR (Fig. 5e). Fig. 12b shows that, for low TR, daily minimum

tide elevation, when channel rip activity is maximized, tends to occur during the patrolled hours in mid-late afternoon when beach attendance (exposure) is maximized. This can explain this disproportionate number of rip related SZIs for TR close to MNTR.

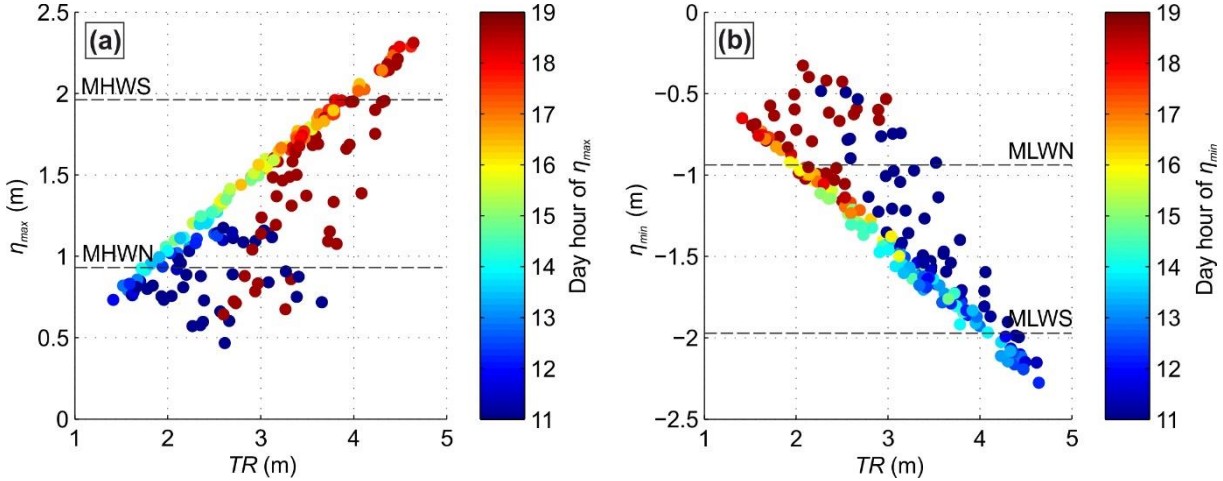

**Figure 12: (a) Daily maximum tide elevation $\eta_{max}$ during lifeguard patrolled hours versus daily tide range TR with colour bar**
**indicating the day hour of $\eta_{max}$ occurrence; (b) daily minimum tide elevation $\eta_{min}$ during lifeguard patrolled hours versus daily tide range TR with colour bar indicating the day hour of $\eta_{min}$ occurrence.**

Our results show that wave energy and tidal elevation are key environmental factors controlling both rip and shore-break related SZIs. Similar observations were made by Scott et al. (2014) for rip rescues along the SW coast of UK, who further developed two composite parameters to determine the mode of rip behaviour and resulting hazard, namely the wave factor

$Wf = HsT02/Hs\acute{T}02$ and the tide factor $Tf = LW - \acute{LW}$, where $LW$ is the daily low water elevation, and $Hs\acute{T}02$ and $\acute{LW}$




are the long-term summer averages. Fig. 13 compares the average background distribution of these two composite environmental parameters against their distributions associated with both rip and shore-break related SZIs. A disproportionate number of rip related SZIs took place on days when low water level was close MLWN (0.25 m < $Tf$ < 0.5 m). Scott et al. (2014) found a disproportionate number of rip rescues for slightly lower water levels, close to MLW (-0.2 m < $Tf$ < 0.2 m). It

is important to note that here we addressed life risk related to rips using reported SZIs, while Scott et al. (2014) normalised rescues (and not necessarily drowning incidents) for beach attendance and therefore addressed rip hazard. Therefore, it is not possible to draw a sharp comparison and instead similarities can be discussed. The slight difference in water level can be explained by evidence that rip activity, and therefore rip hazard, is maximized near mean low tide along the SW coast of UK (Austin et al., 2014), while rips tend to become inactive for water levels near and below MLW in SW France (Castelle et al.,

2006). A disproportionate number of rip related SZIs in SW was also found near MLWS ($Tf$ < -0.5 m, Fig. 13c), which may be the signature of high tide swash rips. This was not observed in Scott et al. (2014) because the upper beaches in SW UK are gently sloping owing to finer beach sand with only little shore-break hazards. A significant peak above the background distribution was observed for 0.6 < $Wf$ < 1.1, that is just below the seasonal average for SW UK (Scott et al., 2014).

In contrast, average or above seasonal average wave conditions ($Wf$ > 0.8) result in a disproportionate number of rip incidents

in SW France (Fig. 3a). A possible explanation is that Scott et al. (2014) addressed rescues in their analysis, while here only drowning incidents are analysed. The $Wf$ patterns found here are, however, similar to those reported by Scott et al. (2014) for mass rescue events. Even clearer thresholds in $Tf$ and $Wf$ are observed for shore-break related SZIs (Fig. 13b,d), but their genericity will need to be addressed at other surf beaches. $Tf$ and $Wf$ therefore appear as powerful composite parameters to address the primary wave and tide control on coast-wide life risks. It is anticipated that similar studies in other settings will

provide more insight into the environmental and social parameters controlling these differences. However, this will require using consistent datasets, that is, addressing the rescues or reported SZIs, and addressing life risk or hazard posed.





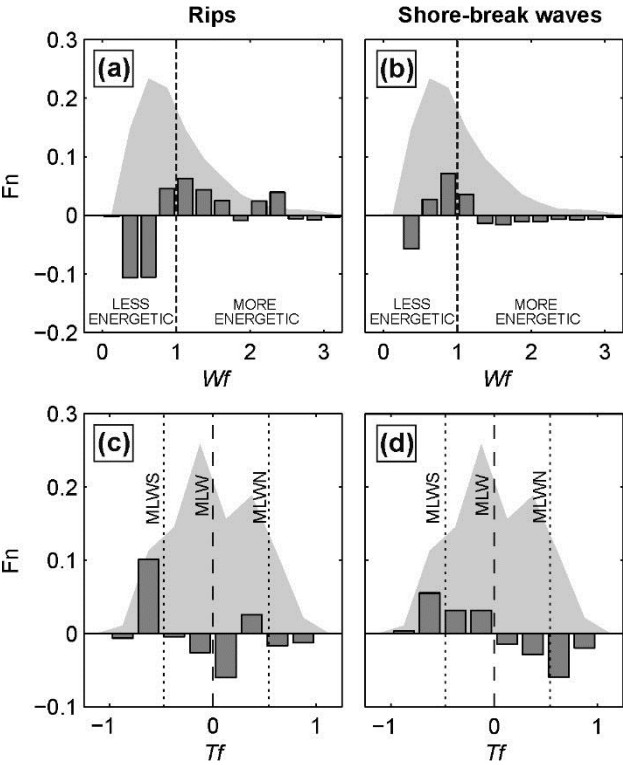

**Figure 13: Normalised frequency distributions Fn of (a,b)** $Wf$ **(wave factor = $HsT\,02/\acute{Hs T}02$) and (b,c)** $Tf$ **(tide factor = $LW - \acute{LW}$ associated with (left-hand panels) rip related injuries and (right-hand panels) shore-break related injuries. Dark grey bars are the difference between the incident-related and the 'average' background frequency distributions, positive values are the proportion of the incident-related distribution greater than the background. In the bottom panels the dashed lines indicate Mean Low Water Spring (MLWS), Mean Low Water (MLW) and Mean Low Water Spring (MLWS).**

As noted in Castelle et al. (2018a), there are large data gaps in the injury report forms used here in this study, including entire lifeguard seasons, making it impossible to robustly analyse interannual variability and temporal trends of the occurrence of SZIs. Fig. 14 displays the percentage of SZIs related to rips, shore-break waves and surfing activity for the 3 summer seasons (2007, 2009 and 2015), which shows strong variations in the occurrence of the different types of SZIs. For example, the summer season of 2015 is characterized by a very small proportion of rip related SZIs (5.1%). In contrast, the 2007 summer is characterized by a low proportion of shore-break related SZIs (38.5%) compared to the two other summers (57.9 and 56.3%). A preliminary analysis does not show substantial differences in overall wave climate that are able to explain this variability. However, it is well established that summer beach morphology in SW France shows strong interannual variability (e.g. Robinet et al., 2016; Castelle et al. 2017), which may explain such variations in SZI distribution.

To further explore this hypothesis, Fig. 15 shows beach profiles surveyed at Truc Vert beach during the three summer seasons examined in this study. It is apparent that the upper beach during the 2007 summer (blue profiles in Fig. 15b) was much more gently sloping (tan $\beta \approx 0.06$ at MHWS) than during the summers of 2009 and 2015 (tan $\beta \approx 0.16$ at MHWS), which result in less shore-break wave hazard and may explain the decreased number of shore-break related SZIs (Fig. 14a). In addition, the





satellite images in Fig. 16 show the presence of deep and complex rip channels incising the inner sandbar systems along the entire coast during the 2007 and 2009 summers (Fig. 16a-b), which were characterised by a large proportion of drowning incidents caused by rips. This contrasts with the reasonably alongshore-uniform sandbar during the summer of 2015 (Fig. 16c), which was not prone to rip related incidents.

5 While long-term morphological data at Truc Vert (Castelle et al., 2017) and satellite images of the coast are available over the last 15 years, such a long-term SZI dataset is challenging to gather. Long-term accurate and sustainable incident reporting of beach related injuries is a global challenge (Williamson, 2006), but it is anticipated that such a dataset combined with long-term beach monitoring programs, including beach survey and large-scale remote sensing will provide new insights into the primary physical and social factors influencing interannual variability of rip and shore-break related SZIs. This will also be a

10 pre-requisite to address the impact on the temporal and spatial occurrence of SZIs in new or improved public safety awareness campaigns on rips and shore-break waves in SW France.

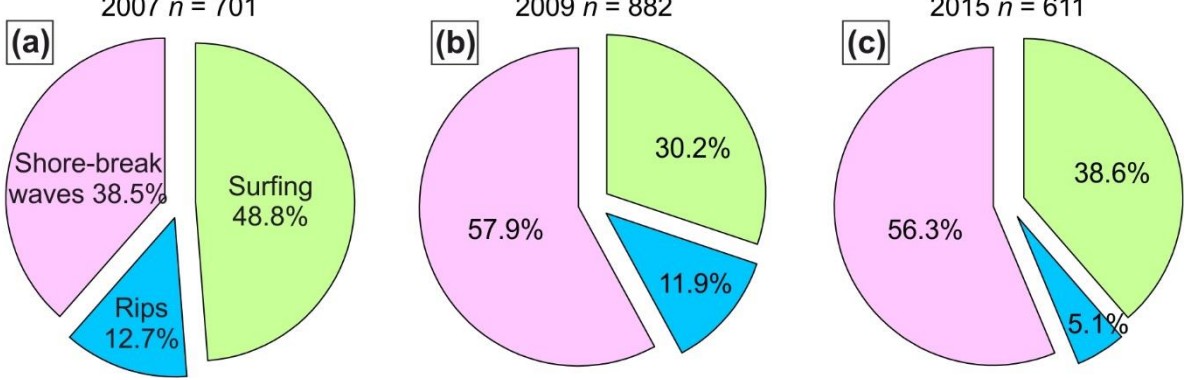

**Figure 14: Percentage of SZIs by year : (a) 2007; (b) 2009; (c) 2015.**




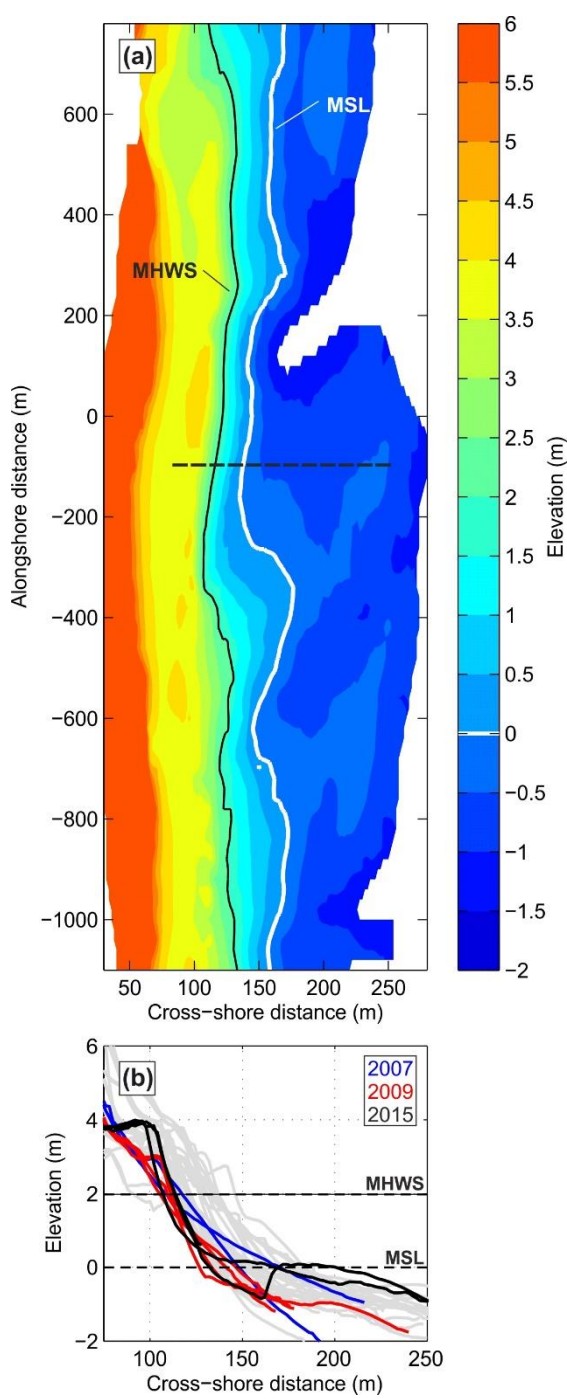

Figure 15: (a) Truc Vert beach morphology surveyed on August 17 2015 with the colour bar indicating elevation in metres. (b) Superimposed alongshore-averaged summer (July-August) beach profiles at Truc Vert beach since 2005 in grey, with the summers of 2007, 2009 and 2015 coloured.



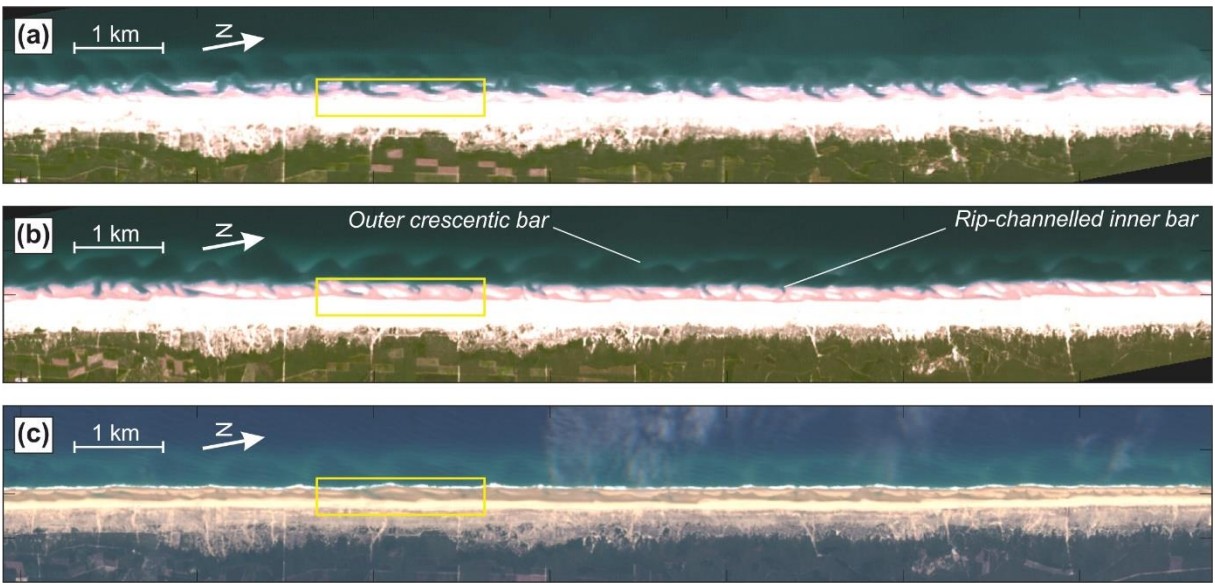

**Figure 16: Satellite images of the Gironde coast around low tide near the vicinity of Truc Vert beach with the survey area in Fig. 15a indicated by the yellow box: (a) SPOT 5 image on June 18 2007; (b) SPOT 5 image on June 24 2009; and (c) Sentinel-2 image on September 24 2015.**

An important result is that the coast-wide highest risk wave conditions for rip related fatal and non-fatal drowning incidents is the presence of a high-energy near shore-normally incident swell waves. High-risk days with very long-period high-energy waves, such as those that occurred on August 5 2007 in SW France (Fig. 7), are consistent with the occurrence of mass rip rescue events observed in similar settings in Scott et al. (2014) in the SW UK. Such long-period high-energy swells can drive large infragravity waves in the surf zone (Bertin et al., 2018), with unsuspecting bathers likely to be swept into the channel at

the passage of the infragravity wave crest. In such conditions, entire groups of bathers can be pulled offshore resulting in mass rescue events (personal communication with SW France lifeguards). Such very long-period near shore-normally incident swells often occur as a result of Tropical Cyclone transitioning into extratropical cyclone in the north Atlantic (Fig. 9). An overall clockwise shift in the angle of wave incidence (more obliquely incident in SW France) and a decrease in wave height is expected in the wave climate of the Bay of Biscay (Charles et al., 2012). But there is also evidence for increased intensity

and frequency of Tropical Cyclone activity in the north Atlantic during the next decades (e.g. Villarini and Vecchi, 2013). Climate change driving increased ocean temperature and an extension of warm weather conditions before and after the summer, may also result in longer tropical cyclone season and extended and higher beachgoer attendance. Collectively, this would result in increased exposure of beachgoers to hazardous very-long period high-energy wave events in SW France and an increase in rips and associated SZIs.

The surfing population of all abilities, including surf schools, was discriminated from regular bathers as only a weak environmental control was found between SZIs and the surfing population. This can be explained by the substantial amount of SZIs reported to occur during a surfing lesson, with training courses primarily provided in the morning and early afternoon regardless of weather and wave conditions. In addition, beginners will prefer low-energy and gentle waves while advanced



surfers will paddle out even for high-energy waves ($Hs > 2$ m) irrespective of weather conditions, pending low or offshore wind conditions, which greatly improve surf-break quality. Overall, more SZIs resulting from surfing activity were reported for low to moderate wave height, large wave period, WNW directional wave incidence, and warm sunny day with low wind independent of tidal stage or range.

Fig. 17 synthesises the primary surf-zone life risks occurring along the sandy beaches of SW France, together with the primary environmental controls. SZIs related to surfing activity can occur throughout the tide cycle when waves break away from the shore (Fig. 17b). High-energy channel rips flow through deeper channels around low tide with the risk maximized for average or above-average wave height, near shore-normal incident, wave conditions and spring tide range (Fig. 17c). Small-scale swash rips flowing through beach cusps located on the upper part of the beach preferably occur at spring high tide (Fig. 17e). Higher

tide levels also see increased shore-break wave activity with incidents disproportionately occurring for just below summer average, near shore-normal incident, wave conditions and spring tide range (Fig. 17d). Tide therefore acts as the most important and complex environmental control on rip and shore-break related SZIs along the meso-macrotidal beaches SW France. While Scott et al. (2014) identified the importance of tides in relation to rip current incidents, the complexity of the tidal control is further evident in this study along the steepest beaches of SW France owing to coarser sand. This result in the presence of

hazardous shore-break waves, high-tide swash rips flowing through beach cusps and surfing-related accidents which all have different wave, tidal and morphological controls. This also suggests that lifeguard management and beachgoer safety along the coast of SW France would benefit from the development of different risk predictors for rip and shore-break wave hazards.





**Figure 17: (a) Primary surf zone life risks along the sandy surf beaches of SW France with: (b) high-risk surfing activity throughout the tide cycle; (c) channel rips primarily flowing between low tide and mid tide and maximized for near shore-normally long period swell; (d) hazardous shore-break waves primarily occurring at high tide on the steepest section of the beach and (e) small-scale swash rips flowing through the centre of beach cusps at spring high tide. The primary environmental controls on beachgoer exposure and surf-zone hazard are synthesised in the top boxes in panel (b-e), with bold arrows, '~' and '?' indicating strong, absent and unknown control, respectively. Wave and morphological controls on swash rip life risk was not addressed here as it was not possible to discriminate from channel rips.**

## 5 Conclusions

The large tide range and energetic, variable, and incident wave climate along surf beaches in SW France results in complex and highly variable surf zone injuries (SZIs) primarily caused by shore-break waves, rip currents and surfing/bodyboarding





related activities. Based on SZIs reported by lifeguards during the summers of 2007, 2009 and 2015, as well as tide and wave hindcast, weather and beach morphology data, SZIs along this section of coast occur disproportionately on warm sunny days with low wind as a result of increased beach attendance and beachgoer exposure to hazards.

When excluding recreational surfers from the analysis, results show that the tide exerts a strong control on both rip and shore-
break related SZIs on beaches in SW France. Shore-break related SZIs occur disproportionately around higher water level, and large tide range, as waves break across the steepest high-tide sections of the beaches. In contrast rip related drownings occur disproportionally at low tide, coinciding with maximum channel rip flow activity. Additional drowning incidents occur at spring high tide, presumably due to small-scale swash rips flowing through the centre of beach cusps. Above seasonal average wave conditions result in a disproportionate number of rip current related incidents in SW France, while more shore-break
related SZIs occur for average or below seasonal mean waves.

Overall, the higher risk wave conditions for rip related fatal and non-fatal drowning incidents is the presence of a high-energy near shore-normal incident swell, with low tide occurring in late afternoon coincident with maximum beach attendance. Beach and surf zone morphology is critical to the hazard posed, with summers with steep upper beach and well-developed inner-bar rip channels characterized by much more shore-break and rip related SZIs, respectively. These results have significant
implications for the future development, or modification, of public education messaging involving interventions targeting beachgoers safety on beaches in SW France.

**Acknowlegments**

We thank the *Compagnies Républicaines de Sécurité* (CRS) headquarters for providing the incident report forms as well as the lifeguard chiefs for filling the incident forms throughout the years. The surf zone injury report forms, which are still used
at the time of writing this paper, were designed by J.M. Campagne in 1999 in close collaboration with CRS lead by Christian Mondon, and civil lifeguards, fire brigade and coastal municipalities. BC was funded by project SONO (ANR-17-CE01-0014) from the *Agence Nationale de la Recherche* (ANR). This study includes the monitoring study site of Truc Vert labelled by the *Service National d'Observation* (SNO) Dynalit (https://www.dynalit.fr), which surveys are financially supported by SNO Dynalit and *Observatoire de la Côte Aquitaine* (OCA, https://www.observatoire-cote-aquitaine.fr). The wave buoy data was
provided by the French Candhis network operated by CEREMA. Weather station and tide gauge data were provided by the Météo France Radome network and the SHOM, respectively. We acknowledge NOAA for providing free access to the tropical cyclone data IBTrACS v3.

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
