# Peer review of "Environmental controls on surf zone injuries on high-energy beaches"

_Natural Hazards and Earth System Sciences, 2019_

## Referee Comment (RC1) · Anonymous Referee #1 · 22 May 2019

Thank you for the opportunity to review this paper. I have previously reviewed the paper whilst it was under consideration elsewhere. In this previous review round, there was no concern on the scientific quality of the paper, merely the scope of the journal. I am pleased to see all suggestions from initial review round have been incorporated. As such, I have no further suggestions, and am able to recommend that the article proceeds to publication.

In the spirit of open discussion and review, I have pasted my initial review comments below to highlight changes made by the authors in response to review: ——————————————

Thank you for the opportunity to review this manuscript, entitled 'Environmental controls on surf zone injuries on high energy beaches'. In the contribution, the authors

attempt to link hydrodynamic forcing information with reports on surf zone injuries. Crucially, the authors aim to extend previous research methodologies, typically focussed on rip current incidents, to provide information on shore break waves and surfing-related injury. I find that the conclusions are largely well supported by the data, and feel that the content is worthy of investigation. I would recommend that the authors address some of the following concerns/comments prior to publication: 1. The reporting of incidents by lifeguards underpins all analysis in this manuscript, but very little information is provided on how lifeguards record incidents. While acknowledging the reference to the earlier paper by Castelle, where I note more detailed information to be given, I would expect to see a more thorough overview in the current contribution. Specifically, how do lifeguards categorize an injury? Is there any validation to their recording of factors, such as shore-breaks? What factors inform whether something is recorded as, for example, a shore break injury? Importantly, is this an open-text field on the report form, or is it categorical and the lifeguards are asked to tick all that apply? 2. The authors place much emphasis on swash rips being the cause for an over-representation of rip related injuries at high tide levels (Fig 5d). In its current form, I do not feel that the authors provide sufficient justification for this conclusion. I note that on line 486 they reference a detailed inspection of reports leading them to conclude that 'some' injuries can be directly attributed to swash rips – but in its current form this is a very weak explanation. What percentage of incidents does this 'some' refer to? No indication provided as to whether this is a majority, or even a significant amount, yet much of the discussion then refers back to this explicit link to swash rips. I agree that this is a very likely explanation for these injuries, but do not feel the authors have yet appropriately justified this link in their manuscript. Minor comments/observations: - Use of T02 in the abstract. This is a non-standard term compared to parameters such as Tp/Tz, and therefore I feel if you wish to use it in the abstract, it should be explicitly defined as mean period. - Ln 62 – The 45 % of rip rescues for the UK seems very low (and the link you provide doesn't work so I am unable to validate it myself). Certainly much other published research on this fact (Scott or Woodward [PhD Thesis: Rip currents in the UK: incident analysis,

public awareness, and education]) suggests a much higher value (Woodward suggests 65%, including the year you cite) – can you justify such a low value here? - Ln 84 – Opening bracket around Lowdon - Fig 6 – both subplots labelled as '(a)' - Fig 8 – state explicitly what you are inferring with the red star - Figure 17 is an important figure in summarising your findings, but I do not find it intuitive in its current form. I still do not understand the direction of the arrows? If you simply wish to infer higher than average, why does the arrow not just point straight up and bold? What does an arrow at 45 degrees infer? Especially when talking about wave period, how can wave period be inferred by a directional arrow? I like the idea of the figure, but feel it needs work in order for it to be instantly interpretable.

---

## Referee Comment (RC2) · Anonymous Referee #2 · 18 Jun 2019

This manuscript investigates the environmental control of surf zone injuries (SZI) along the coast of SW France. To analyze SZI related to shore breaks and rips it builds upon the data set (summers 2007, 2009, and 2015) presented by Castelle et al. 2018a (Natural Hazards). I believed the work present a contribution to the understanding of SZI which has a high social impact. Thus, the work is suitable for the NHESS journal. However, I think that the manuscript requires additional analysis and improvement of the presentation quality before publication. My general and specific comments are provided below:

General comments Exposure.- The results seems to be strongly correlated with exposure (as the authors pointed out) which explains the peaks in the SZI during early August (when most of the people in France are in vacations) and holydays (weekend

[Figure]

of July 14th). Therefore, I think that the environmental factors would be more clearly observed by normalizing the data set by a factor accounting for the number of beach users. I know that analyzing the images from the videocameras might be out of the scope of this work but I guess some statistics about the occupation in coastal cities might be available or testing an existing algorithm is worth to explore. This is something that at least should be more explicitly addressed in the revise ms.

Parameterization.- Wave breaking types can be characterized by parameter relating the wave and beach conditions (e.g., surf similarity, Hunt's). I would expect that the authors explore this and other parameters to be able to extend the current results to other sites. Please check if some of the scatter can be decreased for the shore break analysis.

Additional comments

Page 1, Lines 23, 26, 29, and elsewhere- The word "disproportionately" is employed three times in the abstract and many times in throughout the text. Please find a synonymous to avoid repeating that word too much.

Page 2, Line 13.- It is redundant in this phrase using "annualy" and "each year". Please re-phrase.

Page 4, Line 25.- Fix the text of section 2 (i.e., replace "Introduction" by "Study Area")

Page 5, Line 10.- Fix the numbering of the Figure (i.e. replace "Erreur" by "1").

Page 6, Line 24.- Replace "surf zone Injuries" by "Surf Zone Injuries" Page 6, Line 26.- Replace "Introduction" by "Methods"

Page 7, Line 1.- Replace "Data" by "SZI data"

Page 11-12.- Avoid starting each paragraph with "Fig. . ."

Page 13, line 19-20.- What is the importance of being a Sunday after the national holyday? It looks that could explain such statistics. See my general comment on the

exposure.

Figure 8.- This video system could give a good proxy of beach users by analyzing the images [e.g., Guillen et al., 2008 JCR]. Explore how easy is to employ an algorithm to obtain a time series of the exposure at this site.

Discussion section. - I believe that the role of exposure and parameters that integrate waves and morphology (e.g., surf similarity, Hunt's parameter, etc) must be addressed in this section. Furthermore, the authors should provide recommendation of what additional information should be capture in the forms in the future to provide more insight on the results.

Figure 17.- I would like to see this in terms of parameters instead independent variables (at least for the shore breaks).

---

## Author Response (AR1)

**Reviewer #1: Publish as is**

We thank Reviewer #1 for their support for publication and their constructive comments on a previous version of our manuscript that was submitted to another journal and which, despite positive feedbacks from the 2 reviewers, was declined for publication because it did not satisfactorily fit the scope of the journal.

**Reviewer #2:**

This manuscript investigates the environmental control of surf zone injuries (SZI) along the coast of SW France. To analyze SZI related to shore breaks and rips it builds upon the data set (summers 2007, 2009, and 2015) presented by Castelle et al. 2018A (Natural Hazards). I believed the work present a contribution to the understanding of SZI which has a high social impact. Thus, the work is suitable for the NHESS journal. However, I think that the manuscript requires additional analysis and improvement of the presentation quality before publication. My general and specific comments are provided below:

We thank Reviewer #2 for their support for publication. Below you will see that we did our best to address all their comments. Two of the comments raised by Reviewer #2 could not be addressed rightfully because of the lack of exposure and beach profile data. We hope that we convincingly demonstrate that these two comments cannot be addressed, and we hope that the changes made regarding all the other comments will satisfy the Editor and Reviewer #2 and that our manuscript is now suitable for publication in NHESS.

General comments Exposure.- The results seems to be strongly correlated with exposure (as the authors pointed out) which explains the peaks in the SZI during early August (when most of the people in France are in vacations) and holydays (weekend of July 14th). Therefore, I think that the environmental factors would be more clearly observed by normalizing the data set by a factor accounting for the number of beach users. I know that analyzing the images from the videocameras might be out of the scope of this work but I guess some statistics about the occupation in coastal cities might be available or testing an existing algorithm is worth to explore. This is something that at least should be more explicitly addressed in the revise ms.

We tried different ways to address exposure. It was not possible to use video cameras for this purpose (see detailed response to a later comment and corresponding changes in the text). We tried other ways to address exposure. Statistic on occupation from coastal cities or camp sites is not a good proxy, given that during bad weather conditions tourists are still "occupant" but they are not at the beach, and therefore they do not expose themselves to hazards. Beach attendance and exposure (number of beachgoers in the water) was not indicated in the injury report forms. Note that exposure estimation (number of people in the water) will now be systematically provided in the forms starting this summer after discussion we had with lifeguards earlier this year. The only notable possibility to estimate attendance (but not exposure) is using automated detection of mobile phones passing by beach entries. This has been performed during an entire summer at a given beach by a private company. However, that summer (2017) was not studied here, and the data was not open access anyway. Finally, it must be further highlighted that beach attendance is not necessarily a good proxy of exposure to hazard. In another effort at another site with another dataset (Tellier et al., 2019) and implementing a Bayesian network, one can show that even for high beach attendance, exposure to shorebreak waves is strongly reduced for waves >1.5m as such conditions typically discourage beachgoers to enter the water. Although the absence of attendance/exposure data was already pointed out in our manuscript,

according to this comment by Reviewer #2 we now provide more insight into the lack of exposure data P21 L15-20 (see also additional changes in the reply to comments on the video system): *Statistics* on occupation from coastal cities or camp sites is not a good beach attendance proxy, given that during bad weather conditions tourists do not go to the beach. Finally, it is important to note that beach attendance is not necessarily a good proxy for exposure as, for instance, high surf or cold water temperature can discourage beachgoers to enter the water. This research has highlighted that in order to definitely correlate SZI hazard to environmental parameters, exposure must be quantitatively known. Therefore, a major outcome of this work is that lifeguards will now count people and will systematically provide this information in the new injury report forms.

**Parameterization.- Wave breaking types can be characterized by parameter relating the wave and beach conditions (e.g., surf similarity, Hunt's). I would expect that the authors explore this and other parameters to be able to extend the current results to other sites. Please check if some of the scatter can be decreased for the shore break analysis.**

Once again, this was not possible, given that beach profile data during the studied summer are only available at Truc Vert in the south of Gironde coast, with only 2 surveys per summer, which is not representative of the entire coast nor of the detailed evolution during the summer. Therefore, it was not possible to use the beach profile data, instead tide elevation was considered as a proxy of beach slope. Other parameters have been explored but did not show much improvement and often distracted from the influence of single or existing composite parameters. For instance, given that there is no beach surveys along the entire coast and assuming that beach slope increases with increased beach elevation the parameter  $Bf = \eta T02Hs^{0.5}$  is a good proxy of the surf similarity. The figure below shows the corresponding normalised frequency distributions for shore-break related injuries. Not surprisingly, more shore-break related injuries occur for large Bf, but this does not add much to the existing plots of tide level  $\eta$ , mean wave period T02, and significant wave height Hs.

The fact that other parameters have been tested is now briefly discussed in the revised manuscript P24 L6-11: Other composite parameters were tested for shore-break related SZIs, but none of them provided additional insight into environmental controls. For instance, it is well established that the surf similarity parameter is indicative of breaking wave type, with large values promoting plunging and shore-break waves. Given that beach slope was not available along the entire coast and that one can assume that beach slope increases with elevation up to the berm crest, the modified surf similarity parameter Bf= $\eta T02\sqrt{Hs}$  was tested. Not surprisingly, results (not shown) indicate that more shore-break related injuries occur for large (>5) Bf values.

Figure A. Normalised frequency distributions Fn during the summers of 2007, 2009 and 2015 (light grey region), referred to as 'average' background distribution of  $Bf = \eta T02Hs^{0.5}$  with the dark grey

bars showing the difference between the surfing related and the 'average' background distributions.

**Additional comments**

Page 1, Lines 23, 26, 29, and elsewhere- The word "disproportionately" is employed three times in the abstract and many times in throughout the text. Please find a synonymous to avoid repeating that word too much.

Done, the number of words "disproportionate" and "disproportionately" has been more than halved in the revised manuscript.

Page 2, Line 13.- It is redundant in this phrase using "annualy" and "each year". Please rephrase. Done

Page 4, Line 25.- Fix the text of section 2 (i.e., replace "Introduction" by "Study Area")

Fixed, Section 2 now reads "Regional Setting"

Page 5, Line 10.- Fix the numbering of the Figure (i.e. replace "Erreur" by "1").

Done

**Page 6, Line 24.- Replace "surf zone Injuries" by "Surf Zone Injuries"**

It is now replaced by "SZIs"

**Page 6, Line 26.- Replace "Introduction" by "Methods"**

Section 3 now reads "Data" and comprises subsection "SZI data" and "Environmental data"

**Page 7, Line 1.- Replace "Data" by "SZI data"**

Done

**Page 11-12.- Avoid starting each paragraph with "Fig. . ."**

The four sentences starting with "Fig..." have all been reworded.

**Page 13, line 19-20.- What is the importance of being a Sunday after the national holyday? It looks that could explain such statistics. See my general comment on the exposure.**

We do not think this is important, given that Saturday is week-end anyway. In addition, there is no other Sunday following a national holiday in our time series to further verify if this is important. Contrary to what is happening out of summer holidays, it is important to point out that there is no strong increased exposure during weekends. This can be further shown by the average proportion of SZIs for each weekday, which can be considered as a good proxy of average beach attendance (see Fig. B). This shows that only slightly more SZIs tend to occur on Sundays, while Saturdays are average. This can be explained by most tourists leaving or arriving on Saturdays, and therefore being less likely to go to the beach and expose themselves to hazards.

Figure B. Average number of SZI during weekdays

**Figure 8.- This video system could give a good proxy of beach users by analyzing the images [e.g., Guillen et al., 2008 JCR]. Explore how easy is to employ an algorithm to obtain a time series of the exposure at this site.**

This is a relevant comment and this method of inferring exposure has been explored at the beginning of our project. However, the only decent video monitoring station along this stretch of coast is Biscarrosse, from which the snapshots of Fig. 6 have been taken. It is however not possible to use the method of Guillen et al. or others at this site for two primary reasons: the video station is not satisfactorily maintained and as a result (1) there are many days without images acquired; (2) when the video station was working, during most of the summers only one camera was working which is that looking at the main beach entrance (Fig. C). Given that this stretch of beach is easily packed during sunny days, with most beachgoers spreading along adjacent more remote beaches further north or further south the number of people on the snapshot is not a good proxy of beach attendance. This is however pointed out in our revised manuscript stating that video systems with large coverage could provide relevant information on beach attendance and exposure to hazards P21 L7-15: Although no beachgoer exposure data was available in this study, such environmental conditions are commonly found to contribute to increased beach attendance and beachgoer exposure to hazards (e.g. Ibarra, 2011; Balouin et al., 2014). It is possible to estimate beach attendance using video monitoring systems (e.g. Guillen et al., 2008), however it was not possible to apply such an approach in this study. The Biscarrosse video station (Angnuureng et al., 2017), which is the only suitable video station along this coast, has long periods with no image. In addition, during most of the summers studied here, only one camera was working (Fig. 6). This camera covers a narrow stretch of beach facing the main entry of the coastal resort. This area is rapidly filled on sunny days, with thousands of beachgoers going further north or south along the beach to find some on the dry beach. Accordingly, the number of people from this single camera is not a good proxy for the overall beach attendance.

Figure C. Snapshot of Biscarrosse Beach at the main beach entry on August 4, 2009. The dry beach is packed with holiday-makers in the camera view field. The area is "saturated" and beachgoers all go further south or north along the beach to seek place, which is out of the field of view. Therefore that day the number of people within the camera view field is not a good proxy of the

**Discussion section. - I believe that the role of exposure and parameters that integrate waves and morphology (e.g., surf similarity, Hunt's parameter, etc) must be addressed in this section. Furthermore, the authors should provide recommendation of what additional information should be capture in the forms in the future to provide more insight on the results.**

As discussed above, it is not possible to estimate exposure, and we therefore only added the discussion paragraph given in the response to previous comment. We think that this paper has contributed many novel ideas, including: determining the first robust correlations between multiple SZI types and environmental parameters; identify the role that tidal stage and time of day plays in different SZI; the first connection between interannual morphological and climate relate impacts on SZIs.

**Figure 17.- I would like to see this in terms of parameters instead independent variables (at least for the shore breaks).**

We added the parameters to Fig. 17 but chose to keep the independent variables which, we think, are important to understand the respective contribution of each variable to SZIs and their contribution to hazard and exposure, while parameter Wf for instance accounts for both hazard and exposure.

**Environmental controls on surf zone injuries on high-energy beaches**

Bruno Castelle1,2, Tim Scott3, Rob Brander4, Jak McCarroll3, Arthur Robinet5, Eric Tellier6,7,8, Elias de Korte9, Bruno Simonnet8, Louis-Rachid Salmi6,7,10

[revised manuscript text omitted]